# Cryo-TEM structure of β-glucocerebrosidase in complex with its transporter LIMP-2

Jan Philipp Dobert[1], Jan-Hannes Schäfer [2], Thomas Dal Maso [3,4], Priyadarshini Ravindran[1], Dustin J. E. Huard[5], Eileen Socher [6], Lisa A. Schildmeyer[5], Raquel L. Lieberman [5], Wim Versées [3,4], Arne Moeller [2,7], Friederike Zunke [1,8] ✉ & Philipp Arnold [6,8] ✉

Targeting proteins to their final cellular destination requires transport mechanisms and nearly all lysosomal enzymes reach the lysosome via the mannose-6-phosphate receptor pathway. One of the few known exceptions is the enzyme β-glucocerebrosidase (GCase) that requires the lysosomal integral membrane protein type-2 (LIMP-2) as a proprietary lysosomal transporter. Genetic variations in the GCase encoding gene GBA1 cause Gaucher's disease (GD) and present the highest genetic risk factor to develop Parkinson's disease (PD). Activators targeting GCase emerge as a promising therapeutic approach to treat GD and PD, with pre-clinical and clinical trials ongoing. In this study, we resolve the complex of GCase and LIMP-2 using cryo-electron microscopy with the aid of an engineered LIMP-2 shuttle and two GCase-targeted pro-macrobodies. We identify helix 5 and helix 7 of LIMP-2 to interact with a binding pocket in GCase, forming a mostly hydrophobic interaction interface supported by one essential salt bridge. Understanding the interplay of GCase and LIMP-2 on a structural level is crucial to identify potential activation sites and conceptualizing novel therapeutic approaches targeting GCase. Here, we unveil the protein structure of a mannose-6-phosphate-independent lysosomal transport complex and provide fundamental knowledge for translational clinical research to overcome GD and PD.

Lysosomes are acidic, membrane-engulfed organelles that degrade proteins, glycolipids, and other intra- and extracellular components. A tight interplay of lysosomal membrane proteins and lysosomal hydrolases orchestrate signaling, transport, degradation and the maintenance of an acidic intraluminal pH of 4.5-5.0[1]. Transmembrane proteins are targeted to the lysosome via conventional or non-conventional cell sorting mechanisms depending on C- or N-terminal amino acid tags[2]. Their soluble counterparts require transport proteins

as adapters for lysosomal targeting, mostly relying on the mannose-6-phosphate receptor (M6PR) pathway[3]. Only a few M6PR-independent pathways have been described and the lysosomal hydrolase β-glucocerebrosidase (GCase) is a prominent example for such an M6PR-independent transport mechanism[4]. GCase belongs to the enzymatic family of glycosidases and hydrolyses the glycolipid glucosylceramide (GlcCer) into glucose and ceramide. GCase reaches the lysosome exclusively in complex with its proprietary transport protein

[1]Department of Molecular Neurology, University Hospital Erlangen, Friedrich-Alexander-University Erlangen-Nürnberg (FAU), Erlangen, Germany. [2]Department of Biology/Chemistry, Structural Biology Section, Osnabrück University, Osnabrück, Germany. [3]VIB-VUB Center for Structural Biology, VIB, Pleinlaan 2, Brussels, Belgium. [4]Structural Biology Brussels, Vrije Universiteit Brussel, Pleinlaan 2, Brussels, Belgium. [5]School of Chemistry and Biochemistry, Georgia Institute of Technology, Atlanta, GA, USA. [6]Institute of Functional and Clinical Anatomy, Friedrich-Alexander-Universität Erlangen-Nürnberg (FAU), Erlangen, Germany. [7]Center of Cellular Nanoanalytic Osnabrück (CellNanOs); Osnabrück University, Osnabrück, Germany. [8]These authors contributed equally: Friederike Zunke, Philipp Arnold. ✉e-mail: friederike.zunke@fau.de; philipp.arnold@fau.de

lysosomal integral membrane protein type-2 (LIMP-2), as it lacks mannose 6-phosphorylation[5,6]. LIMP-2 is a type three membrane protein that contains a conventional lysosomal sorting signal (D/EXXXLL(I)) at its C-terminus[7,8]. The GCase/LIMP-2 transport complex forms within the endoplasmic reticulum (ER) and travels through the trans-Golgi network to the lysosome together[5,9]. Other lysosomal proteins (like prosaposin and cathepsins D and H) also utilize M6PR-independent pathways, for example via sortilin or SEZ6L2 (seizure 6-like protein 2), sometimes in addition to M6PR-mediated transport[10–12]. While the structural basis of M6PR-based enzymatic transport has been described in detail[13,14], structural information is missing for atypical lysosomal transport complexes such as GCase and LIMP-2.

Mutations within the GCase-encoding gene GBA1 that impair enzyme function are the monogenic cause for Gaucher's disease (GD). In addition, such mutations are the highest genetic risk factor to develop Parkinson's disease (PD) with at least 3·25% (depending on ethnicity) of PD patients carrying such a GBA1 variants and reduced GCase activity being a common symptom of PD[15,16]. Mutations within the gene encoding LIMP-2 (SCARB2) cause the monogenic Action Myoclonus–Renal Failure syndrome (AMRF) and are investigated as a risk factors for PD[17,18]. Besides its role in GCase transport, LIMP-2 can also travel to the cell surface of epithelial cells and serves as the receptor for Enterovirus 71 and Coxsackieviruses A7, A14 and A16, all causing hand-foot-mouth disease[19,20]. LIMP-2 dimers can occur at the cell surface and function as receptors for phosphatidylserine-rich liposomes there. Upon binding of these liposomes, LIMP-2 targets them to the lysosomal compartement[21]. Within the lysosome, LIMP-2 transports cholesterol from the lysosomal luminal side to the lysosomal membrane through a pore within the luminal domain of LIMP-2[22].

For GD, therapeutic intravenous administration of recombinant GCase is performed as part of an enzyme replacement therapy[23].

For PD, GCase-targeting compounds like ambroxol and BIA 28-6156 are undergoing clinical trials in PD patients to recover compromised GCase activity[24,25]. More activators and chaperones targeting GCase are in development aiming for a disease-modifying treatment of PD in the future[26–28]. Interestingly, LIMP-2 also appears to boost GCase's hydrolytic activity[29]. Here we determine the molecular interaction between GCase and its transporter LIMP-2. This result will advance the design of novel GCase-targeting treatment strategies that do not interfere with LIMP-2 interaction and hence lysosomal transport of GCase.

In this work, we solve the structure of a soluble GCase/LIMP-2 protein complex to 3.7 Å with cryo-EM (cryo-electron microscopy) and single particle analysis. This allows us to develop a near atomic model of the protein complex and the GCase-LIMP-2 interface. We demonstrate the importance of residues identified in the structure through mutational experiments, which leads to the conclusion that a hydrophobic core and a single salt bridge are indispensable for protein-protein interaction. Thus, we identify the key interacting residues that can be exploited in translational approaches to activate GCase in the future.

## Results

### Sample preparation and analysis
To obtain quantitative amounts of the GCase/LIMP-2 protein complex, a soluble LIMP-2 shuttle construct (sLIMP-2) was used[30]. As detailed in our previous study[30], this sLIMP-2 shuttle contains a secretion sequence, which allows it to intercept the physiological transport of GCase to the lysosome and instead actively shuttles it out of the cell. We overexpressed untagged wild type (wt) GCase in a custom stable HEK293F cell line constitutively expressing His-tagged sLIMP-2, leading to the secretion of GCase/sLIMP-2 protein complexes into the cell medium, from which they were purified utilizing Ni-NTA-coupled resin (Fig. 1A). This purification approach eliminated the need to modify

GCase with an affinity-tag and circumvented the need for cell lysis, yielding a sample with high purity while preserving near-native transport and structure of GCase within the complex. To improve physical separation of the protein complex from sLIMP-2 monomers during subsequent size exclusion chromatography (SEC) and to add additional structural features for image processing during single particle reconstruction, two GCase-specific pro-macrobodies (nanobody fused to maltose binding protein, PMb)[31] termed PMb1 and PMb2 were added to the complex after Ni-NTA purification (Fig. 1A). Previous attempts at structural analysis of the GCase/sLIMP-2 complex were unsuccessful in achieving high resolution due to lack of structural features for identification and classification of single particles (low resolution data of GCase/sLIMP-2 complex can be found in Supplementary Fig. 1A, B).

The two PMbs used for this study were created from two nanobodies previously raised against GCase (Nb1 and Nb6 from[26]), both binding at different epitopes[26]. Simultaneous binding of both Nbs to GCase was confirmed via SEC (Supplementary Fig. 1C). Nb1 was included in PMb1 and Nb6 was included into PMb2. SEC of GCase/sLIMP-2 complex after addition of the PMbs revealed an additional peak when comparing samples with and without PMbs, suggesting the presence of a GCase/sLIMP-2/PMb complex as the peak did not match with those for PMb monomers, sLIMP-2 monomers or the sGCase/LIMP-2 complex (Fig. 1B). Western blot analysis of SEC fractions confirmed the presence of sLIMP-2, GCase and both PMbs in fractions corresponding to this novel peak (Fig. 1B, C; fraction 6 (red)). In both cases, with and without PMbs attached, GCase remained enzymatically active when in complex with LIMP-2 (Fig. 1D). Samples from fraction 6 of the SEC were used for negative stain TEM analysis which determined sufficient protein concentration, expected particle size and a homogenous particle morphology (Supplementary Fig. 1D). Consequently, samples were vitrified and submitted to cryo-EM for subsequent single particle analysis.

### Structural overview
Single particle analysis (details in Supplementary Fig. 1E–3) yielded a density map with a global resolution of 3.7 Å with better resolved areas in the center of the structure (Fig. 2A and Supplementary Fig. 3A). Local refinement focusing on GCase and LIMP-2 separately yielded local resolutions of 3.0 Å and 3.1 Å respectively (Supplementary Fig. 2 and 3A). These maps allowed the modeling of one GCase and one sLIMP-2 molecule as well as the nanobody domains of both GCase-bound PMbs (Fig. 2B). The maltose-binding protein domains of both PMbs were only evident as diffuse densities in 2D classes, indicating a substantial degree of flexibility (Supplementary Figs. 4A, B). The protein structure reveals that GCase binds LIMP-2 at the cranial side of LIMP-2. As the PMbs were used to aid in SEC and structural analysis, we will disregard them in the following discussion and focus on the structure of the GCase/LIMP-2 complex alone. Detailed information about the GCase/Nb interaction is included in Supplementary Fig. 4 and Supplementary Table 1 and 2.

### Architecture of the GCase/LIMP-2 complex
GCase binds to the helix 5/7 motif at the cranial side (membrane distant side) of LIMP-2 via a binding pocket formed by its helices 1 and 3 and four loops of the TIM barrel motif (loops A, B, C, D; Fig. 3A and Supplementary Fig. 5). The active site residues of GCase are positioned within the TIM-barrel and contain the catalytically relevant residues Glu235 and Glu340, which are located close to the LIMP-2 binding pocket (Fig. 3B, green residues). The core of LIMP-2 harbors a cholesterol transport channel[22]. The channel opens at the membrane-proximal side and terminates slightly below the cranial part of LIMP-2 (Fig. 3B, right side). The terminal opening of LIMP-2's cholesterol transport channel is located at the opposite side of GCase's active site entry (Fig. 3B). This argues against a role for GlcCer recruitment as GlcCer could not directly be delivered to the active site of GCase. A previous study suggested a role of LIMP-2 as a phosphatidylserine

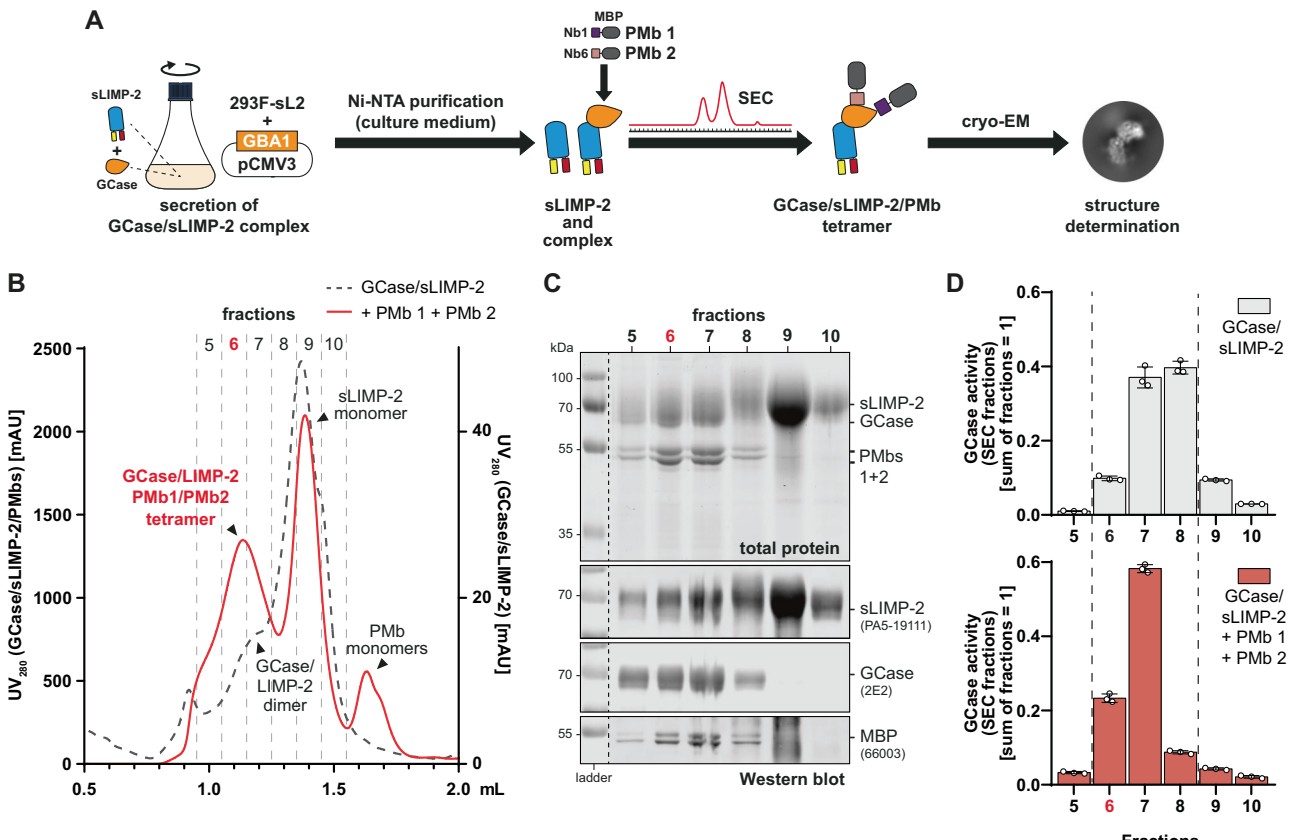

**Fig. 1 | Purification of a GCase/sLIMP-2/PMb tetramer. A** Schematic illustration of protein purification strategy. HEK 293F-sL2 cells were transfected with a GBA1 overexpression plasmid to produce and secrete GCase/LIMP-2 complex. Ni-NTA purification from the culture media yielded sLIMP-2 and GCase/LIMP-2 protein complex. PMbs were added to the purified proteins and SEC was performed to isolate the sGCase/LIMP-2/PMb tetramer, which was then analyzed via cryo-EM. **B** SEC chromatograms of sGCase/LIMP-2 samples with and without PMbs added. Dashed line: no PMbs added. Solid line: PMbs added. Peaks are annotated with corresponding proteins. Addition of PMbs lead to emergence of novel PMb monomer peaks but also caused a shift of the sGCase/LIMP-2 peak, indicating formation of a sGCase/LIMP-2/PMb complex. **C** Analysis of SEC fractions (total protein and Western blot) confirming presence of all four proteins of interest in fractions 5-8, corresponding to the tetramer peak seen in (**B**) (MBP = maltose binding protein as part of the PMbs). Fraction 6 is highlighted in red, as it was used for further structural analyses of the protein complex. **D** Distribution of GCase activity across SEC fractions 5–10 of GCase/sLIMP-2 samples with and without PMbs. As seen in the SEC chromatogram (**B**) and protein analysis (**C**), binding of PMbs leads to a shift of active GCase towards earlier fractions, confirming that GCase retains bioactivity after binding of PMbs (error bars indicates the SD).

receptor by forming a trans-cell homodimer at the cell surface. This dimerization appears blocked while GCase is bound to LIMP-2, as GCase would clash with the second LIMP-2 molecule, and the hypothesized orientation of the LIMP-2 dimer at the membrane would leave no space for GCase to bind (Supplementary Fig. 6A).

Residues lining the active site of GCase engage in substrate binding[32] and were not well resolved, suggesting high flexibility in this region. Crystal structures of GCase show different possible conformations of these loop regions in dependence to crystallization conditions and to bound ligands[32]. Our structure showed no density for a bound substrate within the active site. All four of GCase's annotated N-glycosylation sites (N19, N59, N146, N270) show additional densities representing sugar residues (Supplementary Fig. 7A). LIMP-2 is highly glycosylated and possesses nine annotated N-glycosylation sites (N45, N68, N105, N206, N224, N249, N304, N325, N412), all of which show additional densities corresponding to N-glycosylation (Supplementary Fig. 7B)[33].

For LIMP-2, a pH-dependent conformational switch was described that involves His171 and might induce dissociation of the GCase/LIMP-2 complex once it reaches the lysosome[9]. However, in a previous study, we could show that GCase can still interact with sLIMP-2 at lysosomal pH and that this interaction increases GCase enzymatic activity[30]. We can now allocate His171 near to the complex interface, but apparently, it does not actively participate in the interaction between GCase and LIMP-2 (Fig. 3B). Another binding partner and important regulator of GCase activity inside the lysosome is saposin C (SapC). To further investigate this interaction, we performed crosslinking experiments with recombinant GCase and SapC to obtain information on the binding interface via mass spectrometry. We identified two binding sites, both of which would not interfere with the GCase/LIMP-2 protein complex (Supplementary Fig. 8A, Supplementary Table 3). These predicted binding sites would likely allow binding of SapC and LIMP-2 to GCase simultaneously, which could occur in the lysosome. GCase has been shown to form homodimers in solution and in the presence of an artificial ligand (JZ-4109), which was also beneficial for GCase activity[34]. With regard to the here reported transport complex of GCase and LIMP-2, attachment of a second GCase would be sterically possible as one of the structurally described GCase dimerization sites is not occupied by LIMP-2 (Supplementary Fig. 6B)[35]. However, we have not observed the presence of any GCase dimers alone or GCase dimers attached to sLIMP-2 in our experiments with or without PMbs present (see Fig. 1B–D and Supplementary Fig. 1A-C).

## The GCase/LIMP-2 interface

For detailed molecular analyses of the GCase/LIMP-2 interface, we utilized the Proteins, Interfaces, Structures and Assemblies tool from

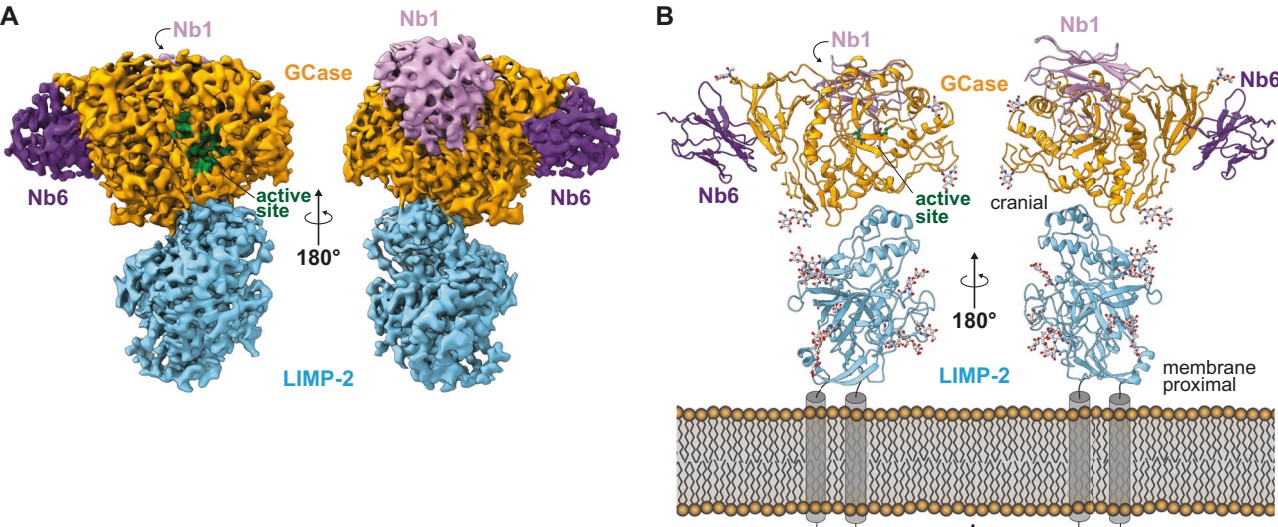

**Fig. 2 | Cryo-EM density map (A) and model (B) of the GCase/sLIMP-2/PMbs tetramer. A** B-factor-sharpened map of electron density obtained from cryo-EM dataset (consensus refinement, resolution: 3.68 Å; B-factor: -152.7 Å²; level: 0.08; dust hidden <11 Å). Surface colored according to the obtained model (see B). **B** Structure of sGCase/LIMP-2/PMb based on the obtained map shown in cartoon style with glycans shown as ball and stick. Blue: sLIMP-2; orange: GCase; green: GCase active site; light purple: Nb1; dark purple: Nb6; white: glycans. The orientation of LIMP-2 at the membrane is illustrated and the N- and C- terminal transmembrane helices present in the full-length protein are indicated (map and model are deposited in PDB as 9FJF and EMDB as EMD-50502).

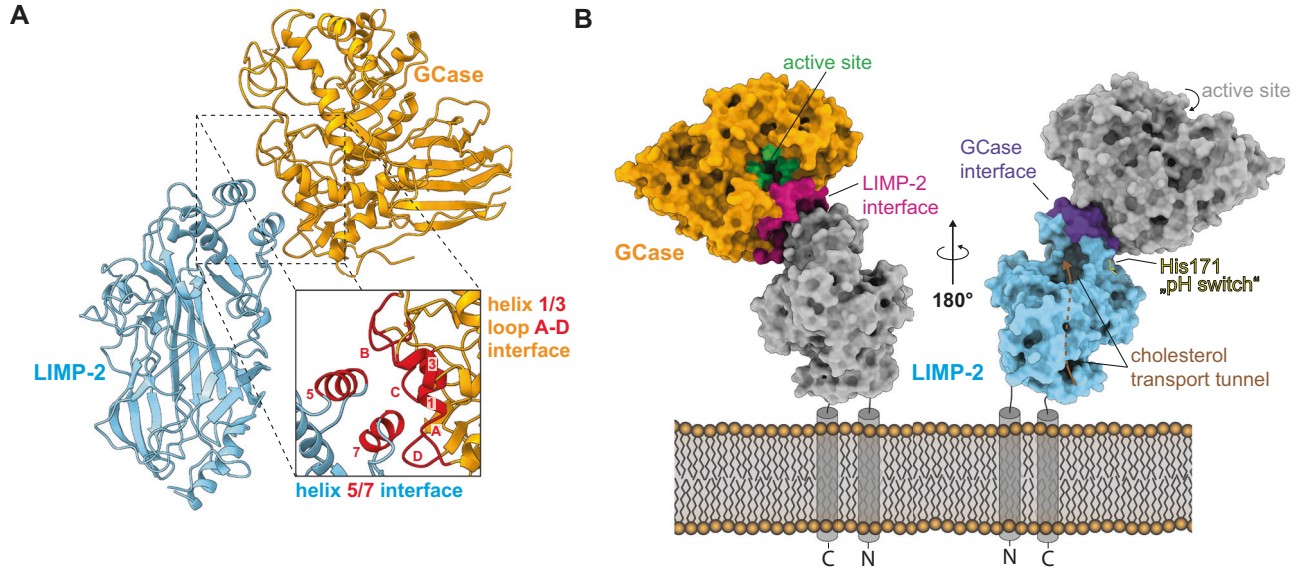

**Fig. 3 | Architecture of the GCase/LIMP-2 complex. A** Model of sLIMP-2 (blue) and GCase (orange) in complex (cartoon style). Magnified view of the binding interface shows helix 5 and 7 on LIMP-2 and helix 1 and 3 as well as loops A, B, C and D on GCase as mediators of interaction (colored in red). **B** GCase/LIMP-2 complex (surface style) with key features and binding interfaces colored and annotated. Left side highlights GCase (orange) with active side cavity (green) and LIMP-2 interface (red). Right side highlights LIMP-2 (blue) with cholesterol transport channel indicated (brown) GCase binding interface (purple) and proposed pH-switch His171 (yellow). The hypothesized orientation of the complex at the membrane is illustrated.

the Protein Data Bank in Europe (PDBePISA)[36], revealing a hydrophobic core shielded by a rim of charged residues (Fig. 4A, center). Within GCase residues Leu91 (helix 1), Leu94, Ala95, Leu96 (loop A), Ile130 (loop B), Leu156 (helix 3), Pro391 (loop C) and Ile406 (loop D) form the hydrophobic core of the interface with LIMP-2 (Fig. 4A, left). These are faced by hydrophobic helix 5 residues Ile155, Ala158, Met159, Ala162, as well as helix 7 residues Leu187, Val190, Phe191 and Pro193 from LIMP-2 (Fig. 4A, right).

In the rim of polar and charged amino acids surrounding the hydrophobic core motif of the interaction interface, hydrogen bonds can form between residues Asn92-Tyr163, Lys155-Phe191,

Arg395-Gln164 (GCase-LIMP-2, Fig. 4B, green). Distances between Asp409-Lys181 and Glu151-Arg192 support salt bridge formation (GCase-LIMP-2, Fig. 4B, yellow; Supplementary Table 4). Interestingly, the binding pattern between LIMP-2 and GCase is similar but not identical to the binding interface between LIMP-2 and Enterovirus71 (EV71, PDB: 6I2K), for which an experimentally determined structure is available[37]. While EV71 shares a hydrogen bond with Tyr163 and a salt bridge with Lys181 of LIMP-2, an additional salt bridge is formed between Lys161 (LIMP-2) and Asp156 (EV71, Supplementary Fig. 6C), instead of Arg192 (LIMP-2) and Glu151 (GCase; Fig. 4B).

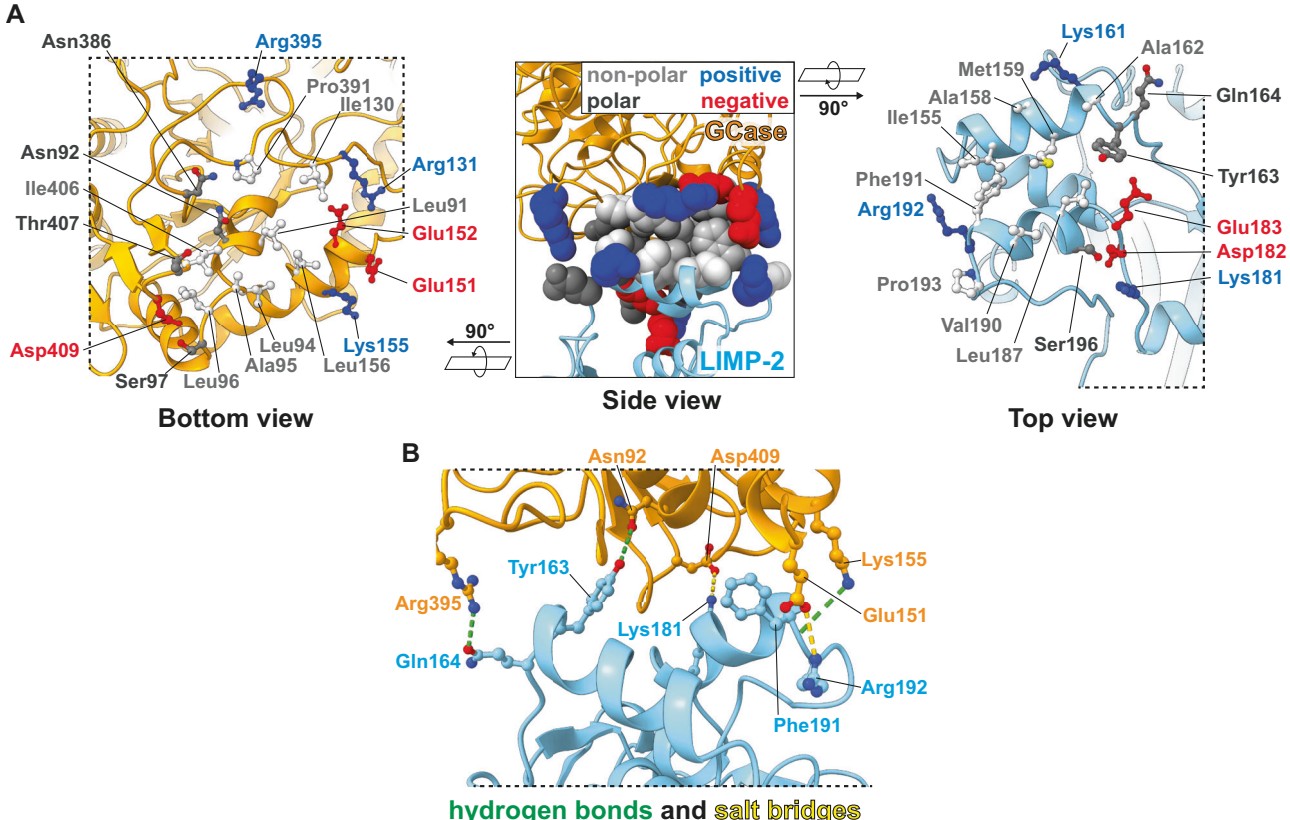

**Fig. 4 | Molecular basis for the GCase/LIMP-2 interaction. A** Side overview (center) and detailed bottom/top views of the GCase/LIMP-2 interface. Interacting residues are shown in sphere style (center) or 'ball and stick' (left/right) and colored according to their polarity and charge. The center of the interaction interface is formed by hydrophobic residues (grey) which are shielded by polar (dark grey) and charged residues (positive: blue, negative: red). **B** Side view of GCase/LIMP-2 (orange/light blue) interface with detailed analysis of hydrogen bonds and salt bridges. Interacting residues are shown as 'ball and stick' model. Atom-atom distances that support formation of hydrogen bonds (green) or salt bridges (yellow) are shown as dashed lines and are detailed in Supplementary Table 1.

## Effect of amino acid changes within the GCase/LIMP-2 interface

All interface residues described for the GCase/LIMP-2 complex are highly conserved among mammalian species (Supplementary Fig. 9A). Our previous mutagenesis experiments in both GCase and LIMP-2 have further shown that the introduction of charged residues into loop A and helix 3 of GCase or helix 5 of LIMP-2 abrogated GCase/LIMP-2 interaction[29,30]. Genetic studies in the context of GD and PD revealed over 300 single nucleotide polymorphisms (SNPs) in the GBA1 gene that lead to single amino acid exchanges in GCase, potentially lowering enzymatic function[38,39]. Although these mutations do not cluster within specific domains, some that are identified as pathogenic or risk-factors reside in the LIMP-2 binding interface (Supplementary Fig. 9B). Thus, we investigated the effects of three central disease-associated amino acid changes (p.E388K, p.R395C and p.D409H) within the interface on the interaction of GCase and LIMP-2 (Fig. 5A). Additionally, on the LIMP-2 side, we selected the previously identified variants p.Y163C to disrupt the hydrogen bond between Tyr163 and Asn92, and p.F191S located within the hydrophobic core of the interface (Fig. 5B). To further assess the importance of the amino acid interactions identified by PDBePISA, we generated the artificial variants p.K181S and p.R192S to disrupt the two identified salt bridges between the complex. Two artificial p.H171A and p.H171K variants previously described by Zachos et al.[9] were also included as controls (full list in Supplementary Table 5). Expression constructs for GCase and sLIMP-2 variants were co-expressed with wt counterparts (wt sLIMP-2 for GCase variants and vice versa) in HEK 293 F cells. Whole-cell lysates were assessed for the expression of variants and activity of GCase, while the impact of variants on complex formation was analyzed via Ni-NTA pulldown

from the cell culture supernatant (as for the wt/wt complex for structural analysis).

For the E388K GCase variant normal expression was detected, but a reduced cell lysate activity (~75%) was measured compared to wt GCase (Fig. 5C, E). The expression of R395C was slightly reduced and cell lysate activity was strongly reduced (50%) compared to the wt GCase (Fig. 5C, E). The D409H variant showed lowest expression in Western blot and enzymatic activity in cell lysates was not different from un-transfected control cells (Fig. 5C, E). Notably, the R395C and D409H variants also lead to reduced expression of wt sLIMP-2, indicating an overall negative effect of these variants on protein expression (Fig. 5C, E). The sLIMP-2 variants Y163C, H171A, K181S and F191S showed reduced expression levels when compared to wt sLIMP-2 (Fig. 5D). Expression of these sLIMP-2 variants together with wt GCase reduced both the expression and activity of wt GCase in whole-cell lysates (Fig. 5C, E). The sLIMP-2 variants H171K and R192S seemed unaffected when compared to wt LIMP-2 in terms of expression and GCase activity (Fig. 5C, E). Ni-NTA pulldown of R395C and D409H GCase was highly reduced when compared to wt or E388K GCase (Fig. 5F). However, yield of wt sLIMP-2 was also reduced for R395C and D409H GCase, suggesting that these variants induce reduced protein expression or trigger a missfolded protein response and target the GCase/LIMP-2 complex towards intracellular degradation (e.g. ERAD pathway; Fig. 5F). Analyzing the K181S LIMP-2 variant reveals a similar picture. This variant interferes with salt bridge formation between Asp409 on GCase and Lys181 from LIMP-2. Both, LIMP-2 and GCase signals are very low in the cell culture supernatant and no GCase activity could be measured after Ni-NTA pulldown (Fig. 5G, H). Thus, the formation of the Asp409-Lys181 salt bridge seems indispensable

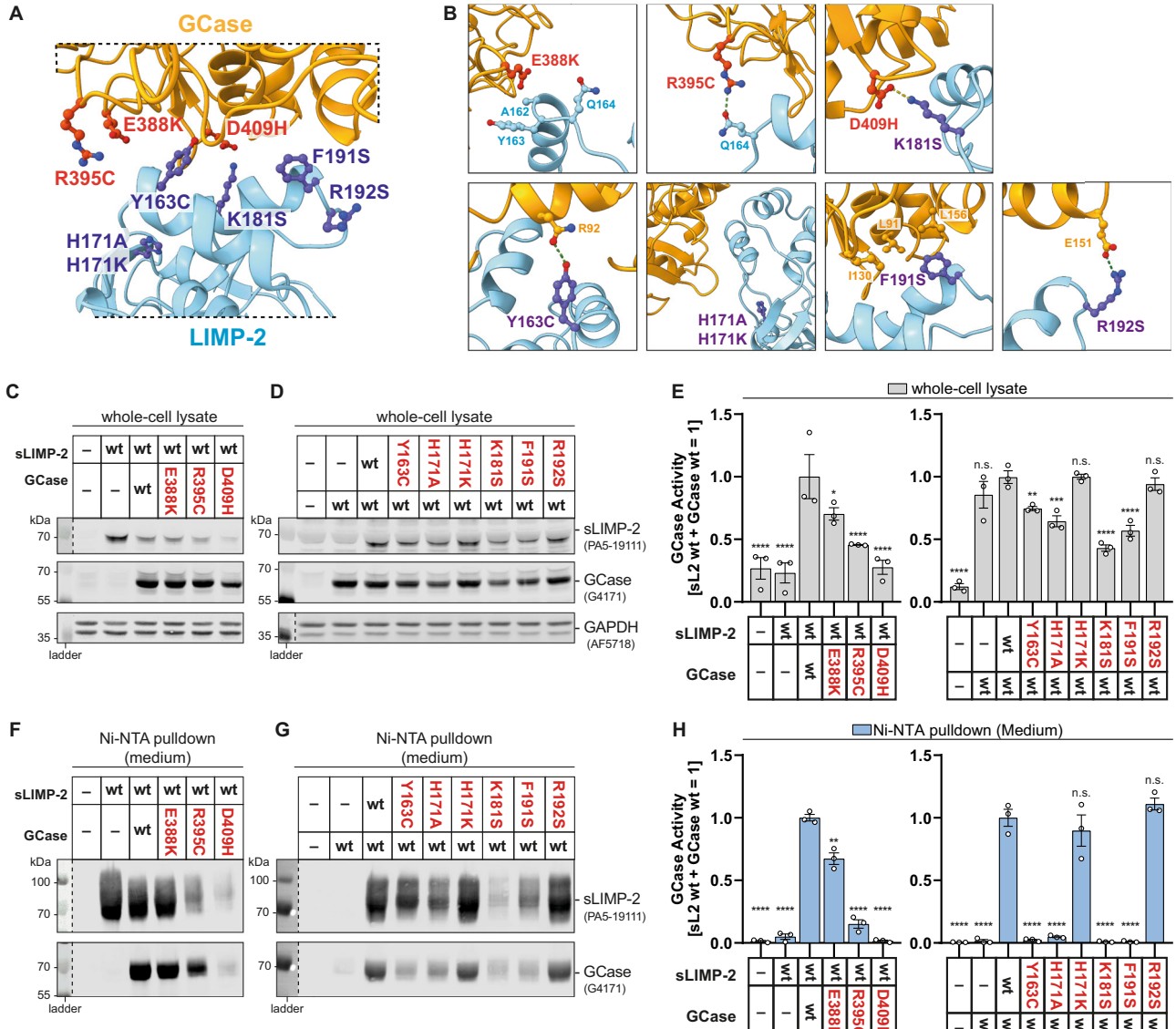

**Fig. 5 | Effect on amino acid substitutions within the GCase/LIMP-2 interface.**
**A** Overview of GCase/LIMP-2 interface (orange: GCase, blue: LIMP-2). Analyzed variants (red: GCase; dark blue: LIMP-2) are highlighted and annotated. **B** Detailed views of each GCase ans LIMP-2 variant generated for this study. The targeted amino acid is highlighted and surrounding or interacting amino acids of the interaction partner are shown and labeled. potential hydrogen bonds (green) and salt bridges (yellow) are shown as a dashed line. **C, D** Western Blot analysis of whole-cell lysates of HEK 293 F cells co-expressing GCase variants with wt sLIMP-2 (C) and sLIMP-2 variants with wt GCase (**D**). All variants were expressed, although some variants showed reduced expression levels and also affected expression of their co-expressed wt interaction partner. **E** GCase activity in whole cell lysate samples from C, D. The GCase variants exhibited reduced activity compared to the wt. Some sLIMP-2 variants negatively impacted GCase activity, corresponding to their effect on GCase expression shown in D (one-way ANOVA, * indicates $p < 0.05$; ** indicates $p < 0.01$; **** indicates $p < 0.0001$, error bars indicate the SD, all compared to wt/

wt). **F, G** Western Blot analysis of Ni-NTA pulldown from culture medium of HEK 293 F cells co-expressing GCase variants with wt sLIMP-2 (**F**) and sLIMP-2 variants with wt GCase (**G**). R395C and D409H GCase variants showed strongly reduced purification yield of both sLIMP-2 and GCase while the E388K variant behaved similar to the wt. **H** GCase activity in Ni-NTA pulldown samples from (**F, G**). Activity of E388K GCase was slightly reduced despite similar-to-wt protein levels observed in F. Activities of R395C and D409H GCase were strongly reduced compared to the wt, corresponding with protein levels. Pulldown samples with LIMP-2 variants Y163C, H171A, K181S and F191S exhibited almost no GCase activity, confirming the low levels of co-purified GCase as shown in (**G**). The H171K sample showed slightly reduced activity and the R192S sample showed similar activity compared to the wt sample, confirming the ability of these variants to bind functional wt GCase (one-way ANOVA, ** indicates $p < 0.01$; **** indicates $p < 0.0001$, error bars indicate the SD, all compared to wt/wt).

for proper protein complex formation. A Lys181-dependent salt bridge also forms between LIMP-2 and its viral ligand EV71 at the cell surface (Supplementary Fig. 6C). For the Y163C and F191S LIMP-2 variants (both located in the interface with GCase) binding of GCase was almost completely abolished as no enzymatic activity could be measured after Ni-NTA pulldown. These results emphasize the importance of an intact hydrophobic interaction motive and the formation of hydrogen bonds between GCase and LIMP-2 for proper complex formation. The R192S

LIMP-2 variant that disrupts the salt bridge between Glu151 from GCase and Arg192 from LIMP-2 shows no expression nor binding defects. Therefore, we conclude that the Glu151-Arg192 salt bridge is dispensable for protein complex formation between GCase and LIMP-2. We also recapitulated the previously reported mutations of His171[9], namely H171A and H171K (Fig. 5D–H). For the H171K LIMP-2 variant, we did not observe a significant change in cell lysate GCase activity when compared to wt (Fig. 5E). GCase activity after Ni-NTA pulldown not

significantly changed (Fig. 5H). In contrast, the H171A LIMP-2 variant showed reduced GCase levels in cell lysates (to 60% of wt) and almost no GCase activity after Ni-NTA pulldown (Fig. 5E, H). Thus, we hypothesize that His171 is vital for proper complex formation (presumably supports proper protein fold), but might not be of vital importance for GCase release in the lysosome. This is further supported by structural comparisons between GCase structures from this study, a pH 4.8 LIMP-2 (PDB: 4TW0) and a pH 7.4 LIMP-2 (PDB: 4TW2)[40]. His171 from the different structures have a very similar conformation and almost no conformational divergence can be observed for neighboring residues (Supplementary Fig. 10).

## Discussion

In this study, we present the structure of an atypical lysosomal transport complex formed by GCase and LIMP-2 in the presence of two pro-macrobodies. This structure enabled analysis of the interaction of both proteins to molecular detail. Helix 5 and helix 7 of LIMP-2 form an interaction motif to engage with a hydrophobic pocket on GCase. While tools such as AlphaFold multimer[41] used for the prediction of structures and protein interactions are getting increasingly accurate and correctly predicted the interacting regions, it was not readily suitable to fully identify interactions between GCase and LIMP-2 on a per-residue level (Supplementary Fig. 11)[41,42]. The interaction interface between both proteins is near the active site of GCase, which might explain the previously published activating effect of LIMP-2 on GCase enzymatic activity[29,30]. The interface is comprised of a hydrophobic core, shielded and stabilized by hydrogen bonds and salt bridges via surrounding hydrophilic residues, which is reminiscent of a typical protein fold organization. Mutagenesis experiments confirmed the importance of the hydrophobic core within the interface and the surrounding hydrophilic residues that can stabilize the interface via hydrogen bonding. Specifically, Tyr163 and Phe191 of LIMP-2 are crucial for GCase binding. Additionally, the salt bridge between Asp409 (GCase) and Lys181 (LIMP-2) seems to be indispensable for proper complex formation. In line, Lys181 and Tyr163 of LIMP-2 engage in binding of EV71 at the cell surface[37]. Binding of LIMP-2 to GCase occurs at a herein identified binding site, which is different from and does not overlap with binding sites suggested for SapC based on our cross-linking data, which in principle would allow simultaneous binding of SapC to the GCase/LMIP-2 complex[43]. Although GCase dimerization is not sterically hindered when in complex with LIMP-2, we did not detect such GCase dimers bound to LIMP-2 in any of our experimental data sets. Importantly, GCase dimer formation is also concentration dependent[44] and we might not have reached the critical concentration of GCase monomers to induce dimer formation. As the structurally solved protein complex of GCase and LIMP-2 derives from an intracellular co-expression strategy, which might lead to higher GCase and LIMP-2 concentrations in the ER than under endogenous conditions, we suggest a predominant 1:1 stoichiometry for the GCase/LIMP-2 protein complex in cell homeostasis. While some studies suggest dissociation of the GCase/LIMP-2 complex upon transport into the acidified lumen of the lysosome[9], other studies showed that LIMP-2 as well as a LIMP-2-derived peptide comprising helix 5 can bind and increase GCase activity under lysosomal conditions[29,30]. The mutational experiments presented here, especially in regards to the previously described pH-switch His171, suggest a more general binding defect for the LIMP-2 H171A variant, which omits further conclusions with regard to dissociation of the complex in the lysosome and thus requires further research. Concerning translational research approaches, the structure of the GCase/LIMP-2 complex fosters two aspects: On the one hand, it allows structure-guided design of GCase activators that capitalize on the GCase/LIMP-2 interaction interface and have to directly act in the lysosome. On the other hand, the GCase/LIMP-2 complex structure facilitates the design of GCase-activating,

cell penetrating compounds that do not interfere with GCase/LIMP-2 complex formation. Targeting of an activator to a specific organelle might be, however, advantageous as it requires less compound and will not flood the organism systemically, which could benefit its side-effect profile. As GCase dysfunction is heavily implicated in both GD and PD, GCase-modulating molecules have emerged as a promising strategy for the treatment of both diseases. Several clinical and pre-clinical studies targeting GCase are ongoing and including knowledge of the GCase/LIMP-2 protein complex structure will improve compound design and selection. Integration of the GCase/LIMP-2 structure into previous and above presented mutational experiments shows that disruption of the hydrophobic core abrogates GCase interaction with LIMP-2[29,30]. The same holds true for the salt bridge formed between Asp 409 from GCase and Lys181 from LIMP-2. These findings might explain the pathogenicity of the GD associated p.D409H p.R395C GCase variants located within the LIMP-2 binding interface[39]. We could show that complex formation of both GCase variants with LIMP-2 is highly impaired, which renders these variants transport deficient and could contribute to their pathology in addition to their general activity impairment[39].

The structure of GCase in complex with LIMP-2 provides details of an M6P-independent lysosomal transport complex. Knowledge of this specific interaction mechanism can be capitalized on for ongoing and future drug design strategies to identify new disease-modifying therapeutic approaches for GD and PD.

## Methods

### Mammalian expression constructs

GCase wt was expressed using a commercially available plasmid (pCMV3-GBA-wt, Sino Biological Inc., Peking, China, #HG12038-UT). The expression construct for the secreted LIMP-2 ectodomain (sLIMP-2) was described previously[30]. The cDNA comprises amino acids 36-431 of LIMP-2 with an N-terminal IgK leader sequence and a C-terminal, TEV-cleavable 10xHis tag in the pCMV3 vector backbone (Sino Biological Inc., Peking, China, #CV011). All plasmids in this study were prepared using an endotoxin free maxi prep kit (Macherey-Nagel, Düren, Germany, # 740424.50).

### Generation of GCase and sLIMP-2 variant expression plasmids

Plasmids for the expression of variants of GCase and sLIMP-2 were generated from their respective wt-encoding plasmids (see previous paragraph). Single or double base pair exchanges were introduced using mutagenesis following the "Round the Horn PCR" protocol by Stephen Floor (https://www.protocols.io/view/around-the-horn-pcr-and-cloning-261geqx7g479/v1). Primers were designed accordingly (Supplementary Table 6) and phosphorylated using T4 polynucleotide kinase (ThermoFisher Scientific, Waltham, MA, United States, #EK0031). The PCR was run using Q5 Polymerase (New England Biolabs, Ipswitch, MA, United States, #E0555S) following manufacturers recommendations (25 cycles) with annealing temperatures calculated using NEBs tm calculator. PCR products were digested with DpnI FD (ThermoFisher Scientific Inc., Waltham, MA, United States, #FD1703) before running on an agarose gel. The PCR product was then extracted from the gel with a GeneJet Gel extraction kit (ThermoFisher Scientific Inc., Waltham, MA, United States, #K0691). Blunt end ligation was carried out over night at room temperature using T4 ligase (ThermoFisher Scientific, Waltham, MA, United States, #EL0011). The ligated product was transformed into XL-1 blue E. coli via heat shock at 42 °C for 45 s and the bacteria were plated on LB-agar containing ampicillin (GCase) or kanamycin (sLIMP-2). Single clone colonies were picked, cultured and plasmid DNA was isolated using a mini prep kit (ThermoFisher Scientific Inc., Waltham, MA, United States, #K0502). Successful cloning of all variant plasmids was confirmed via whole plasmid sequencing (Eurofins Genomics, Ebersberg, Germany).

## Cultivation of HEK 293 F cell lines

HEK 293 F cells (ThermoFisher Scientific Inc., Waltham, MA, United States, #R79007) were cultivated in FreeStyle 293 expression medium (ThermoFisher Scientific Inc., Waltham, MA, United States, #12338018) in vented flat-bottom Erlenmeyer flasks (ThermoFisher Scientific Inc., Waltham, MA, United States, #4115-0125, #4115-0250, #4115-0500) on a MaxQ $CO_2$ Plus orbital shaker (ThermoFisher Scientific Inc., Waltham, MA, United States, #88881102) at 125 rpm, 37 °C and 8% $CO_2$. Flasks were filled to at most 40% of the maximum volume to ensure proper shaking. Regularly, the cell count was determined using a Cellometer Auto T4 Plus cell counter (Nexcelom Biocience LLC, Lawrence, Massachusetts, United States). The cells were maintained at concentrations of $1 \times 10^5$ to $2 \times 10^6$ cells and passaged by diluting the culture with fresh pre-warmed FreeStyle expression medium.

## Transfection of HEK 293 F cell lines

For transfection of HEK 293 F cells, the culture was passaged and diluted to $1 \times 10^6$ cells/mL 24 h before transfection. On the day of transfection, the cells were counted and the desired cell number was transferred to 50 mL tubes. The medium was removed by centrifugation ($500 \times g$, 3 min, RT) and the cells were suspended in fresh FreeStyle expression medium at a concentration of $1 \times 10^6$ cells per mL and vortexed for 10 s. The culture was then returned to the incubator while transfection mix was prepared. For every $1 \times 10^6$ cells in the seeded culture, 1 µg of DNA (per plasmid) and polyethyleneimine (PEI MAX, 1 µg/ml, Polysciences, Inc., Warrington, PA, United States, #24765-1) in a ratio of 1:3 (DNA to PEI) were added to 50 µL (per µg of DNA) of OptiMEM (ThermoFisher Scientific Inc., Waltham, MA, United States, #31985070). The transfection mix was incubated for 25 min at RT before being added to the culture.

## Generation of a stable HEK 293F-sL2 cell line overexpressing sLIMP-2

To increase recombinant protein yield, a HEK 293 F cell line was generated that constitutively overexpresses sLIMP-2. This was achieved by transfecting a 50 mL culture of HEK 293 F cells with the pCMV3-sLIMP-2 plasmid as described in the previous paragraph. 24 h after transfection, 10 mL of culture were taken and centrifuged ($500 \times g$, 3 min, RT) to pellet cells. The medium was removed and the cells were suspended in 10 mL of selection medium made from DMEM (PAN-Biotech, Aidenbach, Germany, #P04-04510) supplemented with 10% fetal calf serum (FCS, PANbiotech, Aidenbach, Germany) and 200 µg/mL hygromycin (Carl Roth, Karlsruhe, Germany, #CP12.2) to select for transfected cells. The cell suspension was then transferred to a standard 10 cm cell dish and the cells were cultured in selection medium as adherent cells without shaking in from this point. The culture medium was renewed twice weekly for a total of 4 weeks to select for stable cells. When confluent, the cells were passaged by detaching with 1 mL of trypsin (PANbiotech, Aidenbach, Germany, #P10-023100) for 5 min and transferring to a new 10 cm dish containing fresh selection medium. After selection, the cells were detached from the confluent plate using 1 ml of trypsin for 5 min. After stopping the trypsin reaction with 4 mL of selection medium and determining the cell count, cells were diluted to 1 cell per 100 µL and single cells were seeded on a 96-well plate for clonal selection. Single cell colonies were continuously cultured in selection medium and passaged first to 24-well dishes and finally to 6-well dishes using trypsin once confluent. Finally, all stable clones were checked for sLIMP-2 overexpression via western blot of whole-cell lysates and the clone showing the highest expression was selected. The clone was then readapted to suspension culture by detaching a confluent 10 cm dish with trypsin and pelleting the cells ($500 \times g$, 3 min, RT) after stopping the reaction (see above). The supernatant was removed and the cells were suspended in 50 mL of pre-warmed FreeStyle medium and transferred to a suspension culture flask. From this point onwards, the culture was cultivated as a

suspension culture as described above. This newly generated stable HEK 293 F line was termed HEK 293F-sL2 and is referred to as such in this manuscript.

## Cell lysis

Cell lysis was used in this study to generate whole-cell lysate samples to analyze the expression of target proteins, either from stable HEK 293F-sL2 cells grown in adherence or transfected HEK 293 F cells grown is suspension. HEK 293F-sL2 cells grown in adherence were washed once with cold PBS, harvested from the plateusing a cell scraper and pelleted ($1000 \times g$, 5 min, 4 °C). HEK 293 F cells in suspension were harvested by transferring th culture to a reaction tube and pelleting the cells ($1000 \times g$, 5 min, 4 °C). After removing the supernatant from the pellets, they were suspended in 50-300 µL of lysis buffer (150 mM phosphate/citrate + 0.25% w/v sodium taurocholate + 0.25% v/v Triton X-100 + 1x proteinase inhibitor cocktail (Roche, Penzberg, Germany, #11697498001, pH: 5.4), followed by incubation on ice for 60 min. After the incubation, the samples were centrifuged ($15,000 \times g$, 15 min, 4 °C) and the supernatant was collected. Protein concentrations were determined using Pierce™ BCA Protein Assay Kit (ThermoFisher Scientific Inc., Waltham, MA, United States, #23227).

## Ni-NTA purification of sLIMP-2/GCase complex from transfected HEK 293F-sL2 cell culture medium

HEK 293F-sL2 cells were seeded and transfected with pCMV3-GBA-wt as described above. 16 h after transfection, a final concentration of 3.75 mM valproic acid (Sigma-Aldrich, St. Louis, MO, United States, #P4543) was added to the culture to increase protein expression. The sLIMP-2 protein construct stably overexpressed by the cell lines binds transiently overexpressed GCase within the cells and is secreted into the cell culture medium. Cultures were harvested 96 h after transfection by centrifugation ($3000 \times g$, 20 min, 4 °C) to remove cells. The conditioned medium containin sLIMP-2 and GCase/sLIMP-2 complex was additionally filtered a 0.22 µm membrane (Carl Roth, Karlsruhe, Germany, #P666.1). Ni-NTA purification of His-tagged proteins was performed at 4 °C using a HisTrap excel 1 mL (Cytiva, Marlborough, MA, United States, #29048586) operated on an ÄKTA pure with sample pump attachment (Cytiva, Marlborough, MA, United States). After column equilibration following manufacturer instructions, the conditioned medium was applied at a flow rate of 0.5 mL/min, leading to binding of sLIMP-2 and GCase/sLIMP-2 to the resin via the His tag on sLIMP-2. Afterwards, the column was washed with 25x CV of NPI-30 buffer (50 mM $NaH_2PO_4$, 300 mM NaCl, 30 mM imidazole (Merck Millipore, Billerica, MA, United States, #814223), pH 7.4) at 1 mL/min. Elution was done with 20x CV of NPI-200 (50 mM $NaH_2PO_4$, 300 mM NaCl, 200 mM imidazole, pH 7.4) at 1 mL/min. Analytical fractions of wash and elution steps were collected. Elution fractions were pooled and concentrated over a 30 K cutoff filter unit (Merck Millipore, Billerica, MA, United States, #UFC5030) for further purification via size exclusion chromatography (SEC).

## Expression, purification and characterization of GCase and sLIMP-2 variants

HEK 293 F cells were cultured as described for the 293F-sL2 line. For transfection, 30 mL cultures were seeded in 125 mL Erlenmeyer flasks at $10^6$ cells/ml. The transfection mix was prepared by adding 30 µg of each plasmid (GCase and sLIMP-2 plasmids or empty vector) and 120 µg PEI to 1200 µL of OptiMEM. The transfection mix was incubated for 25 min at room temperature before addition to the cell culture. Subsequently, cells were cultured for 72 h. At harvest, 1 mL of culture was taken and centrifuged at 1,000xg for 5 min to obtain a medium sample (supernatant). The pellet was lysed in GCase activity buffer to obtain whole-cell lysate. The remaining culture was centrifuged at 4,000xg and 4 °C for 15 min and the clarified conditioned medium was transferred to a 50 mL falcon. A column volume of 100 µL of Ni-NTA

beads (Macherey-Nagel, Düren, Germany, #745400) were equilibrated according to manufacturer's instructions and added to the medium. The mixture was then incubated over-night on a rolling shaker at 4 °C. The next day, the falcons were centrifuged at 500xg and 4 °C for 5 min and the supernatant was discarded. The beads were transferred to a 1.5 mL reaction tube and washed three times with 10x CV 20 mM imidazole wash buffer (50 mM $NaH_2PO_4$, 300 mM NaCl, 20 mM imidazole, pH: 7.4). After, the bound protein was eluted with 2x CV of 400 mM imidazole elution buffer (50 mM $NaH_2PO_4$, 300 mM NaCl, 400 mM imidazole, pH: 7.4) a total of three times and the elution fractions were pooled (600 μL total volume) into the pulldown sample.

For measuring GCase activity, 2.5 μg of whole cell lysate protein (as determined by BCA), 20 μL of supernatant and 10 μL of pulldown sample were used per replicate. For western blot analysis, 10 μg of whole-cell lysate protein and 3 μL of pulldown sample were used per well.

### Size exclusion chromatography (SEC)

SEC was performed to separate sLIMP-2 monomer from co-purified GCase/sLIMP-2 complex, as well as to separate GCase/sLIMP-2/PMb tetramers from sLIMP-2 monomer and PMb monomers. Concentrated protein samples were separated over a Superdex 200 Increase 3.2/300 column (Cytiva, Marlborough, MA, United States, #28990946) using an ÄKTA Pure system (Cytiva, Marlborough, MA, United States). The system tubing and flow path were optimized to reduce void volume. All SEC runs were done at 4 °C with a flow rate of 0.05 mL/min in 20 mM MES, 150 mM NaCl (pH: 7.4). 30 μL of protein solution were injected using a 100 μL loop. Elution fractions of 100 μL were collected.

The dimerization of GCase in the presence of Nb1 and/or Nb6 was assessed using size exclusion chromatography on an AKTA Pure Micro system (Cytiva, Marlborough, MA, United States). Samples were prepared by mixing 20 μM of GCase with 100 μM of either Nb1, Nb6 or a mixture of Nb1 and Nb6. 20 μL of these samples were injected on a Superdex 200 Increase 5/150 GL column (Cytiva, Marlborough, MA, United States) using 10 mM MES, 100 mM NaCl, 1 mM DTT, 5% glycerol pH5.2 as running buffer. All runs were performed at 12 °C with a flow rate of 0.3 mL/min and using a 50 μL injection loop.

### Cloning and purification of GCase-binding Pro-Macrobodies (PMbs)

Nanobodies (Nbs) targeting GCase were generated and characterized in vitro and in cells, as previously described[45]. To aid in structural analysis, a subset of these Nbs (Nbs CA16655 and CA16669, referred to as Nbs1 and Nb6 in this study, encoded in a pHEN29 vector) were converted into larger and rigid scaffolding proteins termed Pro-Macrobodies (PMb1 and PMb2, respectively) using previously published procedures[31,46]. First, PCR fragments corresponding to the open reading frames coding for the Nbs and for maltose binding protein (MPB) were generated, using the pHEN29-Nb (CA16655 and CA16669) and the pBXNPHM3 vectors as templates, respectively, and using the primers 5'-TATATAGCTCTTCTAGTCAGGTGCAGCTGGTGGAGTCTG-3' and 5'-TATATAGCTCTTCTCGGGACGGTG ACCTGGGTCCCCTG-3' for the Nbs, and 5'-TATATAGCTCTTCTCCGCCTCTGGTAATCTGGATT AACGG-3' and 5'-TATATAGCTCTTCTTGCACCCGGAGTCTGCG CGTC TTTC-3' for MBP. The resulting amplicons were mixed in a 1:1 ratio and cloned into the destination expression vector pBXNPHM3 (Addgene #110099; a gift from Prof. Janine Brunner) using FX cloning[47]. The resulting vector hence consists of the Nb-coding region directly linked to a C-terminal MBP via a proline-proline linker and preceded by a pelB leader sequence, a deca-histidine tag, an MBP tag and a 3 C protease cleavage site.

The PMb expression vectors were transformed into MC1061 E. coli cells[48], and cells were grown at 37 °C in terrific broth medium. Protein expression was induced at an $OD_{600} = 0.7$ with 0.02% w/v arabinose (Sigma A3256-100G) for 4 h. Cells were harvested and resuspended in

lysis buffer consisting of 50 mM Tris (pH: 8.0), 150 mM NaCl, 20 mM imidazole, 5 mM $MgCl_2$, 10% v/v glycerol supplemented with 10 μg/mL DNAseI (Deoxyribonuclease I from bovine pancreas, DN25-5G, Sigma-Aldrich, Overijse, Belgium), a protease inhibitor cocktail tablet (Complete, Roche) and 1 mg/mL lysozyme (Sigma L6876-10G). After cell lysis using a cell disruptor system (Constant Systems), the lysate was centrifuged for 45 min at 40,000 × g in a Beckman JA-20 rotor. The clarified supernatant was incubated with 6 mL of Ni-NTA resin (1.5 mL/L of culture). The resin was washed with 150 mM NaCl, 40 mM imidazole (pH: 7.6) and 10% v/v glycerol and the protein was eluted with 150 mM NaCl, 300 mM imidazole (pH: 7.6) and 10% v/v glycerol. The N-terminal pelB-10xHis-MBP was removed by cleavage with 3 C protease during overnight dialysis in 10 mM HEPES (pH: 7.6), 150 mM NaCl, 20 mM imidazole and 10% glycerol. A second IMAC was performed to remove the uncleaved protein. The protein was concentrated before size exclusion chromatography using a Superdex 200 pg column (GE Healthcre) equilibrated with 10 mM HEPES (pH: 7.6), 150 mM NaCl, 20 mM imidazole and 10% v/v glycerol. All fractions were supplemented with 20% v/v glycerol and further concentrated using a 30 kDa concentrator (Amicon Ultra-15, Ultracel-30K, UFC903096, Merck Millipore) before flash freezing and storage at -80 °C.

### Isolation of GCase/sLIMP-2/PMb complex

GCase/sLIMP2 complex was generated and purified as described above. After elution from the HisTrap column and before concentration, the protein concentration of the pooled eluate was determined via $A_{280}$ measurement using a Nanodrop (1 A = 1 mg/mL). Pro-macrobodies PMb1 and PMb2 were added to the eluted protein in a 1:7 w/w ratio (200 μg per PMb for 1,400 μg total eluted protein) and incubated over night at 4 °C. Afterwards, concentration and SEC was performed as described above. Fractions 5 and 6 were selected for EM analysis.

### SDS PAGE and Western blot analysis

SDS PAGE was performed in a Tris-glycine buffer system (resolving buffer: ThermoFisher Scientific Inc., Waltham, MA, United States, #HC2215; stacking buffer: ThermoFisher Scientific Inc., Waltham, MA, United States, #HC2115; running buffer: 25 mM Tris, 192 mM glycine, 1% w/v SDS). Samples were prepared by adding 5X Laemmli buffer (0.3 M Tris-HCl, pH 6.8, 10% w/v SDS, 50% v/v glycerol, 5% v/v β-mercaptoethanol, 5% w/v bromophenol blue) to the samples and heating for 5 min at 95 °C. The samples were then separated over a 10% polyacrylamide gel. Afterwards, the proteins were transferred to a PVDF membrane (Merck Millipore, Billerica, MA, United States, #IPFL00010) via wet blotting. The membrane was blocked in 2% w/v fish gelatin (Merck Millipore, Billerica, MA, United States, #G7041) in Tris-buffered saline (TBS; 100 mM Tris-HCl, 685 mM NaCl, pH 7.5) or in Intercept blocking buffer (LI-COR Biosciences, Lincoln, NE, United States, #927-60001) for 1 h at RT. Primary antibodies (see below for dilution) were applied over night at 4 °C. After incubation, membranes were washed three times with TBS-T (TBS + 0.1% v/v TWEEN-20) before incubation with secondary antibodies for 1 h at RT followed by three more washing steps with TBS-T. Acquisition was done using an Odyssey (LI-COR Biosciences, Lincoln, NE, United States) imaging system. All antibodies were diluted in TBS + 2% w/v fish gelatin + 0.1% v/v TWEEN-20 or Intercept antibody diluent (LI-COR Biosciences, Lincoln, NE, United States, 927-65001). Following antibodies and dilution factors were used for Western blot analysis: Primary: goat anti LIMP-2 (polyclonal, ThermoFisher Scientific Inc., Waltham, MA, United States, #PA5-19111; dilution: 1:1,000), mouse anti GCase (monoclonal, clone E2E, Abnova, Taipeh, Taiwan, #H00002629-M01; dilution: 1:1,000), rabbit anti GCase (polyclonal, Sigma-Aldrich, St. Louis, MO, United States, #G4171; dilution: 1:4,000). Mouse anti MBP (monoclonal, clone 4C6H4, Proteintech, Rosemont, IL, United States, #66003-1-lg; dilution: 1:1,000), goat anti GAPDH (polyclonal, R&D Systems,

Minneapolis, MN, United States, #AF5718); Secondary: IRDye 800CW donkey anti goat (LI-COR Biosciences, Lincoln, NE, United States, #926-32214; dilution: 1:10,000) donkey anti mouse Alexa Fluor 680 (ThermoFisher Scientific Inc., Waltham, MA, United States, #A10038; dilution: 1:10,000), donkey anti rabbit Alexa Fluor 680 (ThermoFisher Scientific Inc., Waltham, MA, United States, #A10043; dilution: 1:10,000).

Quantification of Western blot data was performed in EmpiriaStudio 3.2 (LI-COR Biosciences, Lincoln, NE, United States).

### Staining of total protein in acrylamide gels
Staining of acrylamide gels after SDS PAGE and Western blot was carried out using Coomassie brilliant blue G-250 (CBB, Carl Roth, Karlsruhe, Germany, #9598.1). The staining was performed following the protocol of Dyballa and Metzger[49]. Acquisition of CBB gels was done using the 700 channel of the Odyssey imaging system (see above).

### GCase enzyme activity assays
Activity of GCase-containing samples (recombinant protein, cell lysate, lysosome-enriched fractions, conditioned culture medium) was measured using the synthetic substrate 4-methylumbelliferyl-β-D-glucopyranoside (4MU, Merck Millipore, Billerica, MA, United States, #M3633). Samples were added to the well of a black 96-well Maxisorp plate (ThermoFisher Scientific Inc., Waltham, MA, United States, #437111). The reaction volume was then increased to 100 μL with GC AA buffer (150 mM phosphate/citrate + 0.25% w/v sodium taurocholate + 0.25% v/v Triton X-100) and a final concentration of 1 μM of 4MU substrate was added. After an incubation at 37 °C for 30 min, the reaction was stopped by adding 100 μL of stop solution (0.1 M glycine, pH: 10.4). GCase activity was quantified via fluorescence of released 4-methylumbelliferone (endpoint measurement) using an Infinite 200 Pro (TECAN, Männedorf, Switzerland), SpectraMax Gemini (Molecular Devices, San José, CA, United States) or ClarioStar (BMG LABTECH, Ortenberg, Germany) multiplate reader [$\lambda_{ex}$: 365 nm; $\lambda_{em}$: 445 nm].

For evaluation, all data sets obtained for GCase enzymatic activity were normalized to the total signal sum of all samples within each replicate experiment to correct for overall signal differences between different measurement devices and replicate measurements.

### Cryo-EM sample preparation and data collection
Prior to cryo-electron microscopy (cryo-EM), sample quality was assessed using negative-stain electron microscopy, following our established protocol[50]. Micrographs were acquired manually using a JEM2100plus transmission electron microscope (Jeol) operating at 200 kV and equipped with a Xarosa CMOS (Emsis) camera, employing a nominal magnification of 30,000, corresponding to a pixel size of 3.12 Å per pixel. A representative micrograph and 2D classes obtained from processing in cryoSPARC (v.4.2) are given in Supplementary Fig. 1A and Supplementary Table 7.

For cryo-EM, 0.6 mg/ml of LIMP-2-GCase-PMb was applied to freshly glow-discharged C-flat grids (Protochips; CF-1.2/1.3-3Cu-50). The grids were glow-discharged using a PELCO easiGlow device, undergoing two runs at 15 mA for 45 s. A sample volume of 3 μL was then applied to a grid immediately before plunge freezing. The vitrification process was carried out in a Vitrobot Mark IV (Thermo Fisher) at 100% relative humidity and 4 °C. The dataset was collected using a Glacios microscope (Thermo Fisher) operating at 200 kV and equipped with a Selectris energy filter (Thermo Fisher) set to a slit width of 10 eV. Movies were recorded using a Falcon 4 direct electron detector (Thermo Fisher) at a nominal magnification of 130,000, corresponding to a pixel size of 0.925 Å per pixel. The recorded movies were saved in the electron-event representation (EER) format and captured at a total dose of 50 electrons per Å². Data was automatically collected using the EPU software (v.2.9, Thermo Fisher) with a defocus range of −0.6 to −2.0 μm. A representative micrograph and 2D classes obtained from processing in cryoSPARC (v.4.2) are given in Supplementary Fig. 1B.

### Cryo-EM data processing
The dataset was processed using cryoSPARC (v.4.2), following the processing workflow shown in Supplementary Fig. 2. Movies were preprocessed with patch-based motion correction, patch-based CTF estimation and filtered by the CTF fit estimates using a cutoff at 5 Å. Blob picking and consecutive 2D classification was used to select particles for a preliminary ab-initio model. From this model, templates were created for repicking. After multiple rounds of 2D classification and heterogenous refinement against the ab-initio model, a high-resolution model was generated via non-homogenous refinement. From this refined model, templates were generated again and particles were repicked and classified again to obtain a better range of particle orientations. Particles yielding best-looking 2D classes were selected to refine the map using non-homologous refinement and local refinement jobs focusing on GCase and LIMP-2. The resulting sharpened maps (map_sharp) were used for initial model building. Supplementary Fig. 3 contains additional data regarding map quality. From these maps, a single map was generated using density modification (phenix.resolve_cryo_em) and phenix.combine_focused_maps, which was used for further refinement of the model.

### Model building, refinement and validation
A starting model was created from existing crystal structures for GCase (PDB:6TN1)[51], LIMP-2 (PDB: 4Q4F)[33] and Nb1 (PDB: 9ENA)[45], which were were docked into the density map in ChimeraX (v.1.7.1)[52]. A preliminary structure of Nb2 was generated via homology modeling based on Nb1 using Modeller (v.10.5)[53,54] and also docked into the density. This initial model was then iteratively fitted using a combination of manual fitting in WinCoot (v.0.9.8.7)[55] and automatic refinement via phenix.real_space_refine within Phenix (v.1.19)[56]. Glycans were added using the carbohydrate module in Coot[57]. MES molecules (solvent) were manually added to the final model. Waters were added using phenix.douse. Final validation reports were automatically generated using MolProbity[58] within Phenix. Reports are attached at the end of the supplementary material (Table S4).

### Interaction analysis
Protein interactions were analyzed with the help of PDBePISA[36].

### Expression of SaposinC and GCase
Protein Expression and Purification. The gene coding for Saposin C or SapC (Homo sapiens origin, codon optimized for E. coli expression and cloned into a pET-28a(+) vector with N-terminal His6-tag immediately followed by tobacco etch virus (TEV) protease cleavage site) was synthesized by GenScript. SapC was expressed in either E. coli strains BL21(DE3) or BL21(DE3)pLysS. Cells were cultured in Luria-Bertani broth (BD) or Superior Broth (US Biological) supplemented with 50 mg L-1 kanamycin sulfate (Sigma-Aldrich). Cells were grown at 37 °C with shaking at 225 rpm until an OD600 value of 0.6–0.8 (LB broth) or 1.5 (Superior Broth) was achieved; at this point, the temperature of the incubating shaker was dropped to 18 °C and allowed to equilibrate at this lower temperature for 1 h. SapC expression was then induced with 1 mM isopropyl-β-D-1-thiogalactopyranoside (IPTG, GoldBio), and cells were allowed to continue shaking overnight (~20 h). After expression, cells were harvested via centrifugation (10 min at 4420 × g, and 4 °C) and flash-cooled in liquid nitrogen.

For purification, SapC-expressing cells were slow-thawed on ice and resuspended in Tris wash buffer (50 mM Tris, 0.5 M NaCl at pH 8.0) supplemented with a protease inhibitor cocktail (cOmplete Tablets EDTA-free, Roche) and deoxyribonuclease I (DNase I, Sigma-Aldrich). Cells were lysed with a French Press, and the resulting lysate was

clarified with ultracentrifugation (up to 1 h at 164,391 × g and 4 °C). SapC was isolated from the supernatant using an Akta purification system equipped with a HisTrap HP 5 mL column (Cytiva). The column was equilibrated into Tris wash buffer containing 30 mM imidazole. After supernatant injection and extensive washing, SapC protein was eluted off the column with a gradient of 30–500 mM imidazole. SapC-containing fractions were concentrated with Amicon Ultra 3000 molecular weight cut-off (MWCO) centrifugal devices (MilliporeSigma) and further purified with fractionation on a HiLoad 16/60 Superdex 75 column (Cytiva), again equilibrated into Tris elution buffer. Protein purity was confirmed with Coomassie stained SDS PAGE.

HEK cells (FreeStyle™293 cell line; Invitrogen #R790-07) which had been stably transfected with GCase (gene on a pcDNA3.1(+) vector and designed to have a C-terminal His10-tag separated from GCase by a small spacer and a TEV protease cleavage site) were used. The cells were cultured and expanded according to the manufacturer's instructions. Cells were grown to a density of 1-2 ×106 cells ml-1. A portion of the culture was used to maintain the culture. The remaining culture was centrifuged at 161 × g for 10 min. The supernatant was decanted, glycerol was added (10% final concentration) to it, and the solution was stored at −80 °C.

GCase enzyme was isolated from cell media, which was slow-thawed on ice immediately prior to protein purification. Media (typically ~450 mL) was treated with protease inhibitor cocktail (cOmplete Tablets EDTA-free, Roche) and loaded with a sample pump onto a HisTrap HP 5 mL column (Cytiva) running on an Äkta purification system. The column was pre-equilibrated into and subsequently washed post load with phosphate buffer (50 mM sodium phosphate at pH 8.0 containing 0.3 M NaCl). After wash-out, additional impurities were removed from bound GCase with a wash consisting of the phosphate buffer supplemented with 80 mM imidazole. GCase enzyme was eluted off the column with a gradient from 80-500 mM imidazole in the phosphate buffer. Protein purity was determined by SDS PAGE.

### Cross-linking of SaposinC with GCase

Chemical Cross-Linking. GCase and SapC were buffer exchanged into McIlvaine buffer at pH 6.0 containing 10 mM NaCl using Amicon Ultra 30,000- and 3000 MWCO centrifugal devices, respectively. Protein concentrations were determined by UV 280 nm absorbance. Due to its estimated low value and to improve accuracy, the molar extinction coefficient of SapC was determined by the Molecular Structure Facility at the University of California, Davis to be 7430.5 M-1 cm-1. Protein components were mixed in microcentrifuge tubes such that SapC was in excess; GCase was at 16.3-20.0 μM, and SapC was at 32.5 μM. Some of the cross-linking reactions were performed in activating buffer (the above McIlvaine buffer but additionally containing 0.1% (v/v) Triton X-100 and 0.25% (w/v) sodium taurocholate). Proteins were allowed to co-incubate for 15 min prior to addition of cross-linker. Working stocks of BS3(d0) (Thermo Scientific) and BS3(d4) (ProteoChem) were prepared at 70 mM in McIlvaine buffer and mixed 1:1 immediately prior to initiating reactions. Similarly, DTSSP (Thermo Scientific) stocks were prepared to 70 mM in the McIlvaine buffer. Cross-linker was added to reaction samples to a final concentration of 1 mM, and reactions proceeded for 1 h at room temperature. Cross-linking reactions were quenched by addition of 1:1 reducing Laemmli buffer, and products were boiled for 3 min and analyzed on 4-15% precast gradient gels (Mini-PROTEAN TGX gels, Bio-Rad). Gels were stained with Coomassie blue and visualized with a ChemiDoc MP Imaging System (Bio-Rad).

### Mass spectrometry to identify GCase and SaposinC interaction sites

Mass Spectrometry Coomassie stained bands of interest were carefully excised from the gel and transferred to microcentrifuge tubes. Gel bands were washed thrice with wash buffer containing 50% acetonitrile and 50 mM ammonium bicarbonate. Washed bands were dehydrated

using 100% acetonitrile and then were reduced by incubation in 5 mM TCEP (tris(2-carboxyethyl)phosphine) for 15 min at 55 °C. TCEP was removed, and gel bands were dehydrated again. Reduced gel bands were incubated for 15 min at room temperature in darkness with 20 mM iodoacetamide to alkylate the reduced proteins. (In the case of DTSSP-treated samples, reduction and alkylation steps were skipped.) Dehydrated gel bands were incubated with chymotrypsin (1 μg per band) in 50 μL of digestion buffer (100 mM Tris-HCl (pH 8.0), 2 mM calcium chloride) at 37 °C for 3 h. Digested peptides were eluted from the gel by vigorously vortexing the gel pieces in 5% acetonitrile and 0.1% formic acid followed by 50% acetonitrile. Eluted peptides were dried in speedvac and resuspended in loading buffer (5% acetonitrile and 0.1% formic acid). Volumes of 15 μL peptide mixture were transferred into sample vials and loaded onto an LC (liquid chromatography) autosampler.

An externally calibrated Thermo Exploris 480 (high-resolution electrospray tandem mass spectrometer, Thermo Scientific) was used in conjunction with a Vanquish Neo nano LC System. A 5 μL sample was aspirated into a 50 μL loop and loaded onto the trap column (Thermo μ-Precolumn 5 mm, with nanoViper tubing 30 μM i.d. × 10 cm). The flow rate was set to 300 nL min-1 for separation on the analytical column (EASY-Spray™ PepMap™ Neo UHPLC Column, 50 cm long, 75 micron internal diameter, C18, reverse phase). Mobile phase A was composed of 99.9% H2O (EMD Omni Solvent), and 0.1% formic acid and mobile phase B was composed of 80% acetonitrile and 0.1% formic acid. A 60-minute linear gradient from 5% to 50% B was performed. The LC eluent was directly nanosprayed into the Exploris 480 mass spectrometer. During the chromatographic separation, the Exploris 480 was operated in a data-dependent mode and under direct control of Thermo Excalibur 4.7.69.37 (Thermo Scientific). The MS (mass spectrometry) data were acquired using the following parameters: 20 data-dependent collisional-induced-dissociation (CID) MS/MS scans per full scan (400–2000 m/z) at 120,000 resolutions. MS2 was acquired at 15,000 resolutions. Ions with single charge or charges more than 8 as well as unassigned charge were excluded. An auto dynamic exclusion window was used. All measurements were performed at room temperature.

Resultant Raw files were searched with Proteome Discoverer 3.1 using Sequest™ HT as the search engine using a custom FASTA database (including amino acid sequence of both interacting proteins). A 20-ppm mass tolerance for the parent ion and 0.02 Da mass tolerance for the fragment ion was used. Fixed value PSM (peptide spectral match) validator was used at 5% FDR. XLinkX-PD detect node was used to detect the cross-linked peptides. XLinkX PD validator was used to filter CSMs (crosslink spectral matches) at 5%FDR.

### Comparisons with other structures and binding of Nbs

In this study we used protein structures previously published by other groups for comparison or interpretation. The following structures (Figure reference, PDB ID and authors) were used in this study: GCase/Nb1 (CA16655) complex (Supplementary Fig. 4C; PDB:9ENA; Dal Maso et al.)[45]; Pro-Macrobody (Supplementary Fig. 4E; PDB: 7OMT, Botte et al.)[31]; LIMP-2 dimer (Supplementary Fig. 7A; PDB: 5UPH; Conrad et al.)[21]; GCase dimer (Supplementary Fig. 7B; PDB: 6T13; Benz et al.)[35]; LIMP-2/EV71 complex (Supplementary Fig. 7C; PDB:6I2K; Zhou et al.)[37].

### Structure prediction

AlphaFold2 multimer was applied for the prediction of the human GCase (Q9H227) and Limp-2 (Q14108). The prediction was run locally using AlphaFold v.2.1.0[59] in the multimer-mode (template date set to 2020-05-14).

### Analysis and depiction of pathogenic GCase variants across the GCase protein structure

For Supplementary Fig. 7B and A, GCase structure was rendered with positions that harbor known amino acid variations highlighted and

color-coded. To obtain this data, we used variant datasets from two studies by Hruska et al. and Parlar et al.[60,61]. From the Hruska study, all variants that led to a single amino acid change were extracted. From the Parlar study, the data was filtered to extract all single amino acid change variants with a reported allele frequency >0. Collectively, all amino acid positions in GCase were variants occurred were highlighted and classified according to their reported severity/pathogenicity. Positions of variants classified by the authors as "severe" were colored red. Positions classified as "mild" by the authors and/or "pathogenic" or "likely pathogenic" by the American College off Medical Genetics (ACMG) were colored red. Positions where variants were classified by the authors as "risk factor"were colored pink. Positions with insufficient information about severity or pathogenicity were colored beige. If variants of different severity were reported for the same position, the position was colored according to the variant with highest severity.

### Generation of conservation plots

Conservation plots were created with BioEdit (version 7.2.5) by using multiple sequence alignments generated with the Clustal Omega program (version 1.2.4).

Sequences used as mammalian homologs for human GCase (UniProtKB entry: P04062): Pan troglodytes (UniProtKB entry: Q9BDT0), Pongo abelii (UniProtKB entry: Q5R8E3), Callithrix jacchus (UniProtKB entry: U3DTR8), Otolemur garnettii (UniProtKB entry: H0XZ17), Mus musculus (UniProtKB entry: P17439), Rattus norvegicus (UniProtKB entry: B2RYC9), Oryctolagus cuniculus (UniProtKB entry: G1SD48), Canis lupus familiaris (UniProtKB entry: A0A8I3NJQ3), Felis catus (UniProtKB entry: M3VVL7), Bos taurus (UniProtKB entry: Q2KHZ8), Sus scrofa (UniProtKB entry: Q70KH2), Balaenoptera musculus (UniProtKB entry: A0A8B8YIJ5), Equus caballus (UniProtKB entry: F6WDY8), Myotis lucifugus (UniProtKB entry: G1PMP8), Loxodonta africana (UniProtKB entry: G3SLB1), Vombatus ursinus (UniProtKB entry: A0A4X2L7Y1), Sarcophilus laniarius (UniProtKB entry: G3W179), Monodelphis domestica (UniProtKB entry: F6VKK2), Ornithorhynchus anatinus (UniProtKB entry: A0A6I8PM52)

Sequences used as mammalian homologs for human LIMP-2 (UniProtKB entry: Q14108): Pan troglodytes (UniProtKB entry: H2QPQ5), Pongo abelii (UniProtKB entry: A0A2J8UU80), Callithrix jacchus (UniProtKB entry: U3F7N9), Otolemur garnettii (UniProtKB entry: H0XF61), Mus musculus (UniProtKB entry: O35114), Rattus norvegicus (UniProtKB entry: P27615), Oryctolagus cuniculus (UniProtKB entry: G1SJL6), Canis lupus familiaris (UniProtKB entry: A0A8C0TQJ6), Felis catus (UniProtKB entry: M3VW08), Bos taurus (UniProtKB entry: A0A3S5ZPN6), Sus scrofa (UniProtKB entry: F1RYT3), Balaenoptera musculus (UniProtKB entry: A0A8C0CL88), Equus caballus (UniProtKB entry: A0A3Q2LUF6), Myotis lucifugus (UniProtKB entry: G1Q1G5), Loxodonta africana (UniProtKB entry: G3T2P7), Vombatus ursinus (UniProtKB entry: A0A4X2KSI2), Sarcophilus laniarius (UniProtKB entry: A0A7N4NIW8), Monodelphis domestica (UniProtKB entry: F7ARU5), Ornithorhynchus anatinus (UniProtKB entry: A0A6I8N5I3)

### Data presentation

Figures of density maps and protein structures were rendered with ChimeraX v.1.7.1 and WinCoot v.0.9.8.7[52,55]. Figures were assembled using Adobe Illustrator v.28.4.1.

### Reporting summary

Further information on research design is available in the Nature Portfolio Reporting Summary linked to this article.

## Data availability

Map and model were deposited in the Electron Microscopy Data Bank (EMDB) and the protein databank (PDB). The PDB ID is 9FJF. The respective EMDB IDs are EMD-50936 (consensus map), EMD-50937 (GCase focus map), EMD-50938 (LIMP-2 focus map) and EMD-50502 (combined focus map).

Mass spectrometry data have been deposited to ProteomeXchange and are available under the accession code PXD058056. Source data are provided with this paper.

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

## Acknowledgements

We like to thank Julia Vandrey (Department of Molecular Neurology, University Hospital Erlangen, Erlangen, Germany) for excellent technical assistance, Prof. Janine Brunner and Dr. Stephan Schenck (VIB-VUB Center for Structural Biology) for providing MC1061 cells and pBXNPHM3 vector and Kilian Schnelle (Department of Biology/

Chemistry, Structural Biology section; Osnabrück University; Osnabrück, Germany) for IT support. We wish to acknowledge the core facilities at the Parker H. Petit Institute for Bioengineering and Bioscience at the Georgia Institute of Technology for the use of their shared equipment, services, and expertise. This work was supported by the Michael J Fox Foundation (GRANT ID: MJFF-019371, MJFF-16697 and MJFF-022745 to F.Z. and P.A.; MJFF-17240 and MJFF-020706 to W.V. and MJFF-010193 to R.L.L.). Further, this work was supported by the Interdisciplinary Center for Clinical Research (IZKF) at the University Hospital of the University of Erlangen-Nuremberg (Jochen-Kalden funding program N8 to F.Z.), the Deutsche Forschungsgemeinschaft (DFG, German Research Foundation) GRK2162 grant number: 270949263 (to F.Z.), the Fonds voor Wetenschappelijk Onderzoek (G031324N to W.V) and a Strategic Research Program Financing from the VUB (SRP95 to W.V.). J.H.S. was supported by the Friedrich-Ebert Foundation.

## Author contributions

Experiments: J.P.D., J.H.S, T.D.M., P.R., D.E.H., L.A.S. Data analysis: J.P.D., J.H.S., E.S., P.A. Data presentation: J.P.D., F.Z., P.A. Manuscript: J.P.D., F.Z., P.A. Technical resources and guidance: R.L.L., A.M., W.V., F.Z., P.A. Study design: J.P.D., F.Z., P.A. Funding: R.L.L., W.V., F.Z., P.A.

## Funding

## Competing interests

Authors declare no competing interest.
