## [Transparent Peer Review file · Nature Communications]

Cryo-TEM structure of β -glucocerebrosidase in complex with its transporter LIMP-2

Corresponding Author: Dr Philipp Arnold

Version 0:

Reviewer comments:

Reviewer #1

(Remarks to the Author)

The authors present the cryo-EM structure of the LIMP-2/GCase complex, which is a M6P-independent lysosomal transport complex. The structural basis of this complex was well-analyzed. In particular, the interface of the LIMP-2/GCase complex was examined at the molecular level, providing a solid foundation for the design of drugs targeting GCase-related diseases. The results were well-presented and the entire story was well-organized. However, there are some concerns that must be addressed before publication is possible, as outlined in detail below.

1: The author used PMbs to make a stabilized LIMP-2/GCase complex, trying to facilitate the cryo-EM data processing of the LIMP-2/GCase complex. Nevertheless, it is still challenging to see the details of PMbs due to its flexibility.

Since the LIMP-2/GCase complex has approximately the same size as the current determined part of the LIMP-2/GCase/PMbs complex, have the authors tried to solve the LIMP-2/GCase structure without adding PMbs?

Moreover, since GCase can form a homodimer in the absence of PMbs, and the authors mentioned that the interaction between LIMP-2 and GCase has little steric hindrance on the formation of GCase dimerization. Therefore, the addition of PMbs to form a tetramer complex raises questions as follows: whether PMbs might compete with the second GCase to bind GCase. Can the authors elaborate on why no LIMP-2/GCase complex with a stoichiometric ratio of 1:2 was observed in the final EM results? There may be a possibility of obtaining a LIMP-2/GCase complex with a 1:2 ratio without adding PMbs after further 3D classification. Furthermore, which states of the LIMP-2/GCase complex, formed with a ratio of 1:1 or 1:2, might represent the physiological working state?

2: The authors emphasized the significance of the interface between GCase and LIMP-2 and its potential impact on drug development. They focused on the D409H mutation as a representative case, highlighting its role in explaining a possible mechanism underlying PD. Therefore, it is important to purify the GCase D409H proteins and test their enzymatic activity and binding affinity to LIMP-2.

3: Since the authors emphasize the importance of the interface between LIMP-2 and GCase, are there any mutations of LIMP-2 on the interface that have the same effects as mutations on GCase? which might partially support the idea that drugs targeting the interface between LIMP-2 and GCase might be indeed feasible.

4: Although the activity assay was mentioned in the Materials and Methods session, no corresponding enzyme activity results were demonstrated in the manuscript. The biological activity must be verified before protein structures are determined, ensuring that the solved structure has biological significance.

5: Regarding the Cell lysis session in the Materials and Methods, it would be helpful to provide more details about the composition of the lysis buffer and where the cell lysis is used.

Furthermore, according to the manuscript, the LIMP-2/GCase complex was obtained by enriching the secreted proteins in the cell medium, eliminating the need to disrupt the cells. It is important to update the methods section to provide clarity on this process.

Additionally, since GCCase only reaches the lysosome in complex with LIMP-2, and the authors utilized the secretion system to obtain the complex, it would be beneficial to outline the advantages of this approach for the readers.

6: In Figure S6, the membrane illustration should be adjusted to avoid the misunderstanding that the LIMP-2 dimer was embedded in the membrane.

Reviewer #2

(Remarks to the Author)

Dobert et al report the first structure of acid- β -glucosidase in complex with its transporter LIMP-2 by cryoEM with assistance from MBP-tethered nanobodies. Unusually, acid- β -glucosidase is transported to the lysosome by LIMP-2 and understanding the interaction is an important step in the development of novel treatments for Gaucher Disease and for Parkinson's Disease, both of which are associated with mutations in acid- β -glucosidase that impair ER-to-lysosome trafficking. The description of the complex structure on its own, however, does not make a compelling story. Further experiments are needed to validate the model and extract impactful and broad biological insights. Suggestions for improvement include:

1. Confirming that the MBP-tethered nanobodies are not forcing an unnatural interaction between acid- β -glucosidase and LIMP-2. For example, mutagenesis studies that disrupt the observed interaction, crosslinking experiments that further confirm the observed model, comparison to low resolution reconstructions without the nanobodies.
2. What evidence can be put forth that the complex used for structural characterization reflects one that can be activated at lysosomal pH? Perhaps this data was meant to be included as there are methods for enzymatic assays in the SI, but no results could be found.
3. In terms of the discussion of the complex with respect to biology, what can be done to better support implications of the structural findings? For example, the effect of the D409H acid- β -glucosidase mutation, pH-dependent mechanism of LIMP-2/acid- β -glucosidase dissociation, etc could be experimentally probed.
4. Both LIMP2 and acid- β -glucosidase reside near or at the membrane, and the discussion of the structure in context of the membrane needs to be more clear. What are the two lines of membrane at the top and bottom of Figure S6A? Does Figure S6B make sense in the context of acid- β -glucosidase interaction with the membrane (ie. the N-terminus)?
5. Discussions of dimerization of both proteins should be further supported by additional data. Is the LIMP-2 dimer presented even seen in solution or could it possibly be an artifact of crystallography? Also, the authors claim that "GCCase dimerizes in the presence of SAPC as well as some artificial ligands, which were suggested to be beneficial for GCCase activity." There is also literature precedent for acid- β -glucosidase monomerization in the presence of SapC (Gruschus et al. 2015, *Biochem. Biophys. Res. Commun.* "Dissociation of glucocerebrosidase dimer in solution by its co-factor, saposin C"). Additional support for the role of dimerization (or not) for LIMP2 or acid- β -glucosidase would strengthen the study.

Reviewer #3

(Remarks to the Author)

A biologically-relevant structure of the GCCase-LIMP2 complex would aid in the design of small molecules or peptides or proteins that could be useful for the treatment of Gaucher disease and/or the form of Parkinson's that is linked to GD. However, it is unclear from this paper that the structure presented is biologically relevant. There are several issues that need to be resolved:

1. The critical issue is whether this tetrameric complex is SELECTED due to the affinity of two previously characterized antibodies or whether they are merely tags that allow the structure to be determined. Related is the issue of how the COMBINATION of nanobodies effect the structure of GCCase and its binding to LIMP2. The bioRxiv manuscript that reported the production and characterization of the two nanobodies show potential effects on GCCase activity and GCCase thermal stability...both structure-driven. HOWEVER, the combination of N1 and N6 is NOT reported. This is critical information
2. The binding of saponin C to GCCase is NOT UNDERSTOOD (models have not been tested) at a structural level....so any interpretations of how LIMP-2 binding and sapC binding are related should NOT be included.
3. I would like to see an experimental test of the relevance of the structure (as opposed to a rationalization based on existing data that is itself not compelling). Demonstrate that elimination of a key interaction (the His 171 "switch" for example) can disrupt transport to the lysosome.
4. assay of GCCase activity in the presence of taurocholate and the absence of a membrane or micellar surface are not relevant. Taurocholate is likely an allosteric activator itself!
5. The fact that the active site residues are not well-resolved in the complex argue against allosteric activation as the mechanism of LIMP2 binding

I strongly feel that the impact of this structure will be minimal if presented in its current form BUT would be significant if bolstered with a few mutagenesis experiments....

Version 1:

Reviewer comments:

Reviewer #1

(Remarks to the Author)

The authors have adequately addressed my comments and concerns.

Reviewer #3

(Remarks to the Author)

I am very happy (and impressed) with the additional data and discussion added by the authors. This kind of response is unfortunately rare. I am in favor of publication

Reviewer #4

(Remarks to the Author)

I have gone over the MS by Dobert et al. and find that the authors have done a commendable job in addressing the concerns raised by the three first-round reviewers. The new data on Saposin C are suggestive but not compelling, but this is included in the Supplementary info and does not affect the main paper. The authors state that the complex with SaposinC will be addressed in future studies, and I agree. The MS is well written and well presented. The central conclusions are sound and bolstered by the new mutagenesis data.

Rebuttal letter

REVIEWER COMMENTS

Reviewer #1 (Remarks to the Author):

The authors present the cryo-EM structure of the LIMP-2/GCase complex, which is a M6P-independent lysosomal transport complex. The structural basis of this complex was well-analyzed. In particular, the interface of the LIMP-2/GCase complex was examined at the molecular level, providing a solid foundation for the design of drugs targeting GCase-related diseases. The results were well-presented and the entire story was well-organized. However, there are some concerns that must be addressed before publication is possible, as outlined in detail below.

The authors thank the reviewer for the overall positive comments and we hope that our additional experiments and explanations will convince you to suggest publication of the manuscript.

1: The author used PMbs to make a stabilized LIMP-2/GCase complex, trying to facilitate the cryo-EM data processing of the LIMP-2/GCase complex. Nevertheless, it is still challenging to see the details of PMbs due to its flexibility.

Since the LIMP-2/GCase complex has approximately the same size as the current determined part of the LIMP-2/GCase/PMbs complex, have the authors tried to solve the LIMP-2/GCase structure without adding PMbs?

The authors thank the reviewer for this comment and addition of the PMbs helped to overcome two major problems that we encountered before their utilization.

1. The complex without PMbs runs very close to the LIMP-2 monomers on SEC and only forms a shoulder that is not truly separated from the LIMP-2 monomer peak. This is visible in the SEC chromatograms depicted in Figure 1B (in MS and below). Addition of the PMbs helped us in obtaining a sufficiently pure complex sample for cryo-EM.

Figure 1B: GCase/LIMP-2 protein complex runs very close to the LIMP-2 monomer peak (grey dashed line). After addition of the two PMbs the GCase/LIMP-2/PMbs complex is separated from the LIMP-2 monomers

2. We faced problems with correct alignment of particles during single particle analysis of sLIMP-2/GCase samples before. The GCase/sLIMP-2 protein complex has very little low-resolution symmetry-breaking features for particle alignment. Due to this, we were unable to obtain high-resolution 2D classes from initial particle sets. The PMbs added enough features visible at low resolution to the single particles to overcome this problem. In 2D class sum images the maltose binding protein part of the PMbs can be seen (FigureS4A). In the final 3D

reconstruction the maltose binding protein is not visible as the conformation is too flexible and only the nanobody parts of the PMbs can be resolved.

Figure S4A: Class sum images from the GCCase/LIMP-2/PMb1/PMb2 protein complex. Red arrowheads indicate the maltose binding protein part of the PMbs.

To show our experimental progress, we have included preliminary data from negative-stain analysis of previous GCCase/sLIMP-2 dimer sample without PMbs, also to address the question regarding stoichiometry further down. Rigid docking of the final atomic model of GCCase/LIMP-2 obtained from our cryo-TEM data reveals a good accordance in size and shape with the obtained low-resolution structure. The data can be found as new supplement data in Figure S1A, B.

Figure S1A, B: A) negative stain electron microscopic image and class sum images from the GCCase/LIMP-2 protein complex without PMbs. B) 3D map of GCCase/LIMP-2 reconstructed from negative stain images (upper row). Rigid docking of the atomic model of GCCase and LIMP-2 as reconstructed from the cryo-TEM experiments (lower row).

Moreover, since GCCase can form a homodimer in the absence of PMbs, and the authors mentioned that the interaction between LIMP-2 and GCCase has little steric hindrance on the formation of GCCase dimerization. Therefore, the addition of PMbs to form a tetramer complex raises questions as follows:

whether PMbs might compete with the second GCCase to bind GCCase. Can the authors elaborate on why no LIMP-2/GCCase complex with a stoichiometric ratio of 1:2 was observed in the final EM results? There may be a possibility of obtaining a LIMP-2/GCCase complex with a 1:2 ratio without adding PMbs after further 3D classification. Furthermore, which states of the LIMP-2/GCCase complex, formed with a ratio of 1:1 or 1:2, might represent the physiological working state?

We thank the reviewer for this insightful comment, which is in line with comments from the other reviewers. We hope that we can provide sufficient experimental data to convince you that the initial complex present in the cell culture supernatant is in a 1:1 stoichiometry and that the PMbs do not

alter the stoichiometric ratio. But of course, we cannot fully rule out that under different conditions, other stoichiometric ratios might occur.

First of all, we now included negative stain data on the GCCase/LIMP-2 complex without PMBs. The resulting 3D map corresponds in size and shape to a 1:1 stoichiometric protein complex (Figure S1A, B).

Second, by the design of our purification approach, all purified GCCase must be bound to sLIMP-2. Therefore, we expect all GCCase detected in SEC fractions to be part of a GCCase/sLIMP-2 complex. Even without the presence of PMBs, we observe only a single peak in the SEC chromatogram that contains GCCase, suggesting that only one stoichiometry is present for the GCCase/LIMP-2 complex. Elution of the complex peak close to the sLIMP-2 monomer peak during SEC suggest a particle size increase that is more likely corresponding to a single GCCase molecule. If the initially purified complex would have 2:1 stoichiometry, competitive binding of the PMBs would yield free GCCase as an additional peak in the chromatogram, which we did not observe. In line with another review comment, we have also added GCCase activity data in SEC fractions (new Figure 1D), confirming that A: GCCase in the purified complex is bioactive and B: no free GCCase can be detected after PMBs addition.

Figure 1B, D: B) SEC chromatograms of sGCCase/LIMP-2 samples with and without PMBs added. Dashed line: no PMBs added. Solid line: PMBs added. Peaks are annotated with corresponding proteins. Addition of PMBs lead to emergence of novel PMb monomer peaks but also caused a shift of the sGCCase/LIMP-2 peak, indicating formation of a sGCCase/LIMP-2/PMb complex. D) Distribution of GCCase activity across SEC fractions 5-10 of GCCase/sLIMP-2 samples with and without PMBs. as seen in the SEC chromatogram (B) and protein analysis (C), binding of PMBs leads to a shift of active GCCase towards earlier fractions, confirming that GCCase retains bioactivity after binding of PMBs.

In an earlier manuscript (Dobert et al., 2024; PMID: 38666485), we expressed GCCase alone or in combination with LIMP-2 and performed SEC experiments. GCCase alone (Figure 3E, F; gray dashed line) runs in earlier fractions than co-expressed with LIMP-2. Together with data from another paper that indicates GCCase dimer formation in a concentration dependent manner, we might not reach the critical concentration for GCCase monomers to form dimers when co expressed together with LIMP-2 (Gruschus JM et al., Lee JC. Dissociation of glucocerebrosidase dimer in solution by its co-factor, saposin C. *Biochem Biophys Res Commun.* 2015 Feb 20;457(4)). Additionally, we cannot exclude an impact of the His-tag attached to the GCCase when expressed alone. For endogenous GCCase and GCCase co-expressed with LIMP-2 no prominent GCCase peak could be detected in the same fractions.

Figure 3 E, F from Dobert et al., 2024: E) SEC and F) Western blot analysis of GCCase/LIMP-2 (blue line), LIMP-2 alone (black line) and His-tagged GCCase alone (gray dashed line).

We additionally performed analytical SEC experiments with commercially available GCCase (Velaglucerase) in the presence or absence of one or both Nbs (from the PMBs). Within the given buffer system (10 mM MES, 100 mM NaCl, 1 mM DTT, 5% glycerol pH5.2) and a concentration 10-times above the K_D reported for dimer formation, we did not observe GCCase dimers. However, we observed a clear Nb dependent shift, demonstrating binding of the Nbs. We included this data as new Figure S1C. Additionally, we also assessed the activity for GCCase alone and in the presence of both Nbs and while the activating effect of individual Nbs was demonstrated previously, we can now show that addition of both Nbs, has no adverse effects for GCCase activity.

Figure S1C and additional data: C) SEC analysis of GCCase alone (blue), Nb1 (yellow) and Nb6 (green), as well as a combination of GCCase with individual Nbs (Nb1 red, Nb6 orange) or both (black). Additional data) activity assay of GCCase alone or in combination with one or two Nbs. Condurotol- β -epoxide (CBE) was used as a negative control.

Overall, we hope we can convince the reviewers that the GCCase/sLIMP-2 complex with a 1:1 stoichiometry represents the intracellular transport state and that binding of the PMBs does not interfere with the stoichiometry of the complex obtained using our purification strategy. Even in strong overexpression of GCCase and LIMP-2, that should lead to some crowding of GCCase in the ER, we did not observe any other stoichiometry than the reported 1:1. We also adapted our manuscript and emphasized this aspect with more detail.

2: The authors emphasized the significance of the interface between GCCase and LIMP-2 and its potential impact on drug development. They focused on the D409H mutation as a representative case, highlighting its role in explaining a possible mechanism underlying PD. Therefore, it is important to purify the GCCase D409H proteins and test their enzymatic activity and binding affinity to LIMP-2.

We thank the reviewer for this crucial suggestion, which is in line with the next comment and other reviewer's suggestions in regards to analysis of GCase and LIMP-2 variants in the interface, their effects on complex formation and their role in pathology.

To address these comments, we have generated a total of three GCase and six sLIMP-2 variants, for which we performed experiments to assess expression, activity and capability to form a GCase/LIMP-2 complex. Along the previously mentioned D409H variant, we included two more disease-associated GCase variants in the interface region (E388K and R395C). We further generated two SNP-derived sLIMP-2 variants (Y163C and F191S), as well as four artificial sLIMP-2 variants to target amino acid interactions identified by our structural analysis (H171A, H171K, K181S and R192S). The K181S variant targets the salt bridge formed with Asp409 and is thus complimentary to the D409H variant. All variants were co-expressed in HEK 293F cells along their respective wt interaction partner. We obtained whole-cell-lysates to assess expression and activity of the variants and performed Ni-NTA pulldown experiments from the cell culture supernatant to purify GCase/sLIMP-2 variant complexes via the LIMP-2 attached 10xHis-tag.

We have put together this data in a new Figure 5 (see below) and added a new chapter to our manuscript presenting our findings. We think that these experiments significantly substantiate and improve our study and thank the reviewer again for this thoughtful suggestion.

3: Since the authors emphasize the importance of the interface between LIMP-2 and GCase, are there any mutations of LIMP-2 on the interface that have the same effects as mutations on GCase? which might partially support the idea that drugs targeting the interface between LIMP-2 and GCase might be indeed feasible.

In contrast to GCase, where the pathological impact of many variants is known, the impact of LIMP-2 variants, especially on interaction with GCase, is less clear. The role of LIMP-2 variants in PD is still debatable, with SNPs considered to be possible PD risk factors located in non-coding parts of the gene (Hopfner et al., 2013; PMID: 23408458 and Usenko et al., 2021; PMID: 33227372). There are LIMP-2 variants that are associated with action myoclonus renal failure (AMRF), but the role of GCase dysfunction in this disease is not fully clear. Still, to address this point, we have included two LIMP-2 variants (Y163C and F191S) to our variant analysis experiments (see previous comment) that were found as SNPs and may have pathogenic potential in the context of AMRF. Both variants are located within the GCase-binding interface and resulted in impaired expression of sLIMP-2 and almost abolished GCase-binding. It was suggested before that, parts of AMRF pathology are caused by GCase dysfunction (Coluci et al., 2024; PMID: 39512127). Our newly included experimental data would strengthen this suggestion.

Figure 5: Effect on amino acid substitutions within the GCCase/LIMP-2 interface. **A:** Overview of GCCase/LIMP-2 interface (orange: GCCase, blue: LIMP-2). Analyzed variants (red: GCCase; dark blue: LIMP-2) are highlighted and annotated. **B:** Detailed views of each GCCase and LIMP-2 variant generated for this study. The targeted amino acid is highlighted and surrounding or interacting amino acids of the interaction partner are shown and labeled. potential hydrogen bonds (green) and salt bridges (yellow) are shown as a dashed line. **C, D:** Western Blot analysis of whole-cell lysates of HEK 293F cells co-expressing GCCase variants with wt sLIMP-2 (C) and sLIMP-2 variants with wt GCCase (D). All variants were expressed, although some variants showed reduced expression levels and also affected expression of their co-expressed wt interaction partner. **E:** GCCase activity in whole cell lysate samples from C, D. The GCCase variants exhibited reduced activity compared to the wt. Some sLIMP-2 variants negatively impacted GCCase activity, corresponding to their effect on GCCase expression shown in D. **F, G:** Western Blot analysis of Ni-NTA pulldown from culture medium of HEK 293F cells co-expressing GCCase variants with wt sLIMP-2 (F) and sLIMP-2 variants with wt GCCase (G). R395C and D409H GCCase variants showed strongly reduced purification yield of both sLIMP-2 and GCCase while the E388K variant behaved similar to the wt. **H:** GCCase activity in Ni-NTA pulldown samples from F, G. Activity of E388K GCCase was slightly reduced despite similar-to-wt protein levels observed in F. Activities of R395C and D409H GCCase were strongly reduced compared to the wt, corresponding with protein levels. Pulldown samples with LIMP-2 variants Y163C, H171A, K181S and F191S exhibited almost no GCCase activity, confirming the low levels of co-purified GCCase as shown in G. The H171K sample showed slightly reduced activity and the R192S sample showed similar activity compared to the wt sample, confirming these variants ability to bind functional wt GCCase.

4: Although the activity assay was mentioned in the Materials and Methods session, no corresponding enzyme activity results were demonstrated in the manuscript. The biological activity must be verified before protein structures are determined, ensuring that the solved structure has biological significance.

We apologize for this oversight as we intended to include the activity data in the original manuscript's supplement. We have now added the activity data for the GCCase/sLIMP-2 complex to the main manuscript instead (Figure 1D), as we agree that it is majorly important to validate and show bioactivity of purified proteins before structural analysis. Activity of GCCase could be confirmed after purification in complex with sLIMP-2, both with and without addition of PMbs. We hope that our inclusion of activity data supports the biological relevance of the obtained structure (see Figure above).

5: Regarding the Cell lysis session in the Materials and Methods, it would be helpful to provide more details about the composition of the lysis buffer and where the cell lysis is used.

We have added more detailed information to the cell lysis section. We originally used cell lysis only for analysis of stable HEK 293F-sL2 clones. In the revised manuscript, we added analysis of GCCase and sLIMP-2 variants, for which we used the same lysis protocol to obtain whole-cell lysate samples. We modified the manuscript and supplemental material accordingly.

Furthermore, according to the manuscript, the LIMP-2/GCCase complex was obtained by enriching the secreted proteins in the cell medium, eliminating the need to disrupt the cells. It is important to update the methods section to provide clarity on this process.

We have updated the methods section of our manuscript to better clarify or exact purification process. We have also added a little more detail to the purification strategy in the main text of the manuscript.

Additionally, since GCCase only reaches the lysosome in complex with LIMP-2, and the authors utilized the secretion system to obtain the complex, it would be beneficial to outline the advantages of this approach for the readers.

We have added a more detailed description of our purification strategy to the main text of our manuscript and also mentioned the benefits of this approach, circumventing the need for cell lysis (improved purity) and tagging of GCCase (retention of native structure).

6: In Figure S6, the membrane illustration should be adjusted to avoid the misunderstanding that the LIMP-2 dimer was embedded in the membrane.

We apologize for the potentially confusion illustration and have adjusted the figure S6 to better convey that the indicated membrane position is only hypothesized. We have further modified the reference to this figure in the main manuscript to avoid misinterpretation.

Reviewer #2 (Remarks to the Author):

Dobert et al report the first structure of acid- β -glucosidase in complex with its transporter LIMP-2 by cryoEM with assistance from MBP-tethered nanobodies. Unusually, acid- β -glucosidase is transported to the lysosome by LIMP-2 and understanding the interaction is an important step in the development of novel treatments for Gaucher Disease and for Parkinson's Disease, both of which are associated with mutations in acid- β -glucosidase that impair ER-to-lysosome trafficking. The description of the complex structure on its own, however, does not make a compelling story. Further experiments are needed to validate the model and extract impactful and broad biological insights. Suggestions for improvement include:

1. Confirming that the MBP-tethered nanobodies are not forcing an unnatural interaction between acid- β -glucosidase and LIMP-2. For example, mutagenesis studies that disrupt the observed interaction, crosslinking experiments that further confirm the observed model, comparison to low resolution reconstructions without the nanobodies.

We thank the reviewer for this comprehensive list of suggestions to improve the robustness of our findings. Concerns with regard to the stoichiometry of the complex are in line with other reviewers and we have included additional data and performed new experiments to address this matter.

1. The GCCase/sLIMP-2 complex was purified before addition of PMBs and eluted as a single SEC peak (Figure 1B, C, D), which shifted after the addition of the PMBs. This shows that the formation of the GCCase/sLIMP-2 was independent from PMBs and the binding of PMBs to the complex did not force a novel interaction between GCCase and sLIMP-2. The interaction of GCCase and LIMP-2, as well as helix 5 specifically, was also further characterized in our previous study (Dobert et al.).
2. The obtained GCCase/LIMP-2 complex shows enzymatic activity before and after the addition of the PMBs. Thus, PMBs do not alter functionality of the protein complex (new Figure 1D).

Figure 1B-D in revised manuscript: B: SEC chromatograms of sGCCase/LIMP-2 samples with and without PMBs added. Dashed line: no PMBs added. Solid line: PMBs added. Peaks are annotated with corresponding proteins. Addition of PMBs lead to emergence of novel Pmb monomer peaks but also caused a shift of the sGCCase/LIMP-2 peak, indicating formation of a sGCCase/LIMP-2/Pmb complex. C: Analysis of SEC fractions (total protein and western blot) confirming presence of all four proteins of interest in fractions 5-8, corresponding to the tetramer peak seen in B (MBP = maltose binding protein as part of the PMBs). Fraction 6 is highlighted in red as it was used for further structural analyses of the protein complex. D: Distribution of GCCase activity across SEC fractions 5-10 of GCCase/sLIMP-2 samples with and without PMBs. as seen in the SEC chromatogram (B) and protein analysis (C), binding of PMBs leads to a shift of active GCCase towards earlier fractions, confirming that GCCase retains bioactivity after binding of PMBs.

3. We now provide additional low resolution structural data from our study of the GCCase/sLIMP-2 complex without nanobodies present. Although we could not obtain a high-resolution structure without the aid of the PMbs, we did perform negative-stain EM analysis and obtained a low-resolution reconstruction of the GCCase/sLIMP-2 complex. This low-resolution density map allows rigid docking of the atomic model deduced from the cryo-map of GCCase/sLIMP-2 without PMbs. Negative stain data did not deliver any other complex than 1:1 stoichiometry. which indicates that our experimental setup indicating that the PMbs do not significantly alter the interaction and orientation of GCCase and LIMP-2 in complex.

Figure S1A, B: A) negative stain electron microscopic image and class sum images from the GCCase/sLIMP-2 protein complex without PMbs. B) 3D map of GCCase/sLIMP-2 reconstructed from negative stain images (upper row). Rigid docking of the atomic model of GCCase and LIMP-2 as reconstructed from the cryo-TEM experiments (lower row).

We have conducted extensive mutagenesis experiments targeting the interaction interface from both sides, GCCase and LIMP-2. For this, we generated disease-associated variants located within the interfaces of both proteins, as well as artificial variants to specifically disrupt the amino acid interactions identified in our structure model. All variants were analyzed in terms of expression, GCCase activity and the ability to form a GCCase/sLIMP-2 complex without the presence of PMbs. We could show that most of the generated variants impaired or almost abolished formation of a complex, showing that the amino acid interactions in the protein structure of the GCCase/sLIMP-2 complex with PMbs bound reflects the physiological state of the GCCase/sLIMP-2 complex, at least with regard to the interface (see Figure below comment 3).

2. What evidence can be put forth that the complex used for structural characterization reflects one that can be activated at lysosomal pH? Perhaps this data was meant to be included as there are methods for enzymatic assays in the SI, but no results could be found.

We thank the reviewer for this comment, which is partially in line with other reviewers. We have included GCCase activity measurements after purification in complex with sLIMP-2 and after addition of the PMbs to Figure 1D of the main manuscript. These activity measurements are conducted at pH 5.4, confirming that the GCCase in the obtained complex is active at lysosomal pH, also after addition of the PMbs.

3. In terms of the discussion of the complex with respect to biology, what can be done to better support implications of the structural findings? For example, the effect of the D409H acid-b-

glucosidase mutation, pH-dependent mechanism of LIMP-2/ acid- β -glucosidase dissociation, etc could be experimentally probed.

We thank the reviewer for this crucial suggestion, which is in line with the next comment and other reviewer's suggestions with regards to analyses of GCase and LIMP-2 variants in the interface, their effects on complex formation and their role in pathology.

To address these comments, we have generated a total of three GCase and six sLIMP-2 variants, for which we performed experiments to assess expression, activity and capability to form a GCase/LIMP-2 complex. Along the previously mentioned D409H variant, we included two more disease-associated GCase variants in the interface region (E388K and R395C). We further generated two SNP-derived sLIMP-2 variants (Y163C and F191S), as well as four artificial sLIMP-2 variants to target amino acid interactions identified by our structural analysis (H171A, H171K, K181S and R192S). The K181S variant targets the salt bridge formed with Asp409 from GCase and is thus complementary to the D409H variant. All variants were co-expressed in HEK 293F cells along their respective wt interaction partner. We obtained whole-cell-lysates to assess expression and activity of the variants and performed Ni-NTA pulldown experiments from the cell culture supernatant to purify GCase/sLIMP-2 variant complexes via the LIMP-2 attached 10xHis-tag.

To address the specific question on pH-induced dissociation, we included the previously reported mutations for His171 of LIMP-2 (Zachos et al., 2012; PMID: 22537104). We can now report, that His171 is most likely important for proper folding of the LIMP-2 interaction interface with GCase. Even at neutral pH we did not observe complex formation when we changed His171 to alanine (H171A). In case we included a positively charged and more bulky lysine, GCase activity was reduced to ~60% after Ni-NTA pulldown. Thus, we cannot make a statement on any pH induced changes of the protein complex and this question needs to be answered in a more comprehensive follow-up investigation, which should also account for other lysosomal interaction partners such as saposin C.

We have put together this data in a new Figure 5 and added a new chapter to our manuscript presenting our findings. We think that these experiments significantly substantiate and improve our study and thank the reviewer again for this thoughtful suggestion.

Figure 5: Effect on amino acid substitutions within the GCCase/LIMP-2 interface. **A:** Overview of GCCase/LIMP-2 interface (orange: GCCase, blue: LIMP-2). Analyzed variants (red: GCCase; dark blue: LIMP-2) are highlighted and annotated. **B:** Detailed views of each GCCase and LIMP-2 variant generated for this study. The targeted amino acid is highlighted and surrounding or interacting amino acids of the interaction partner are shown and labeled. potential hydrogen bonds (green) and salt bridges (yellow) are shown as a dashed line. **C, D:** Western Blot analysis of whole-cell lysates of HEK 293F cells co-expressing GCCase variants with wt sLIMP-2 (C) and sLIMP-2 variants with wt GCCase (D). All variants were expressed, although some variants showed reduced expression levels and also affected expression of their co-expressed wt interaction partner. **E:** GCCase activity in whole cell lysate samples from C, D. The GCCase variants exhibited reduced activity compared to the wt. Some sLIMP-2 variants negatively impacted GCCase activity, corresponding to their effect on GCCase expression shown in D. **F, G:** Western Blot analysis of Ni-NTA pull-down from culture medium of HEK 293F cells co-expressing GCCase variants with wt sLIMP-2 (F) and sLIMP-2 variants with wt GCCase (G). R395C and D409H GCCase variants showed strongly reduced purification yield of both sLIMP-2 and GCCase while the E388K variant behaved similar to the wt. **H:** GCCase activity in Ni-NTA pull-down samples from F, G. Activity of E388K GCCase was slightly reduced despite similar-to-wt protein levels observed in F. Activities of R395C and D409H GCCase were strongly reduced compared to the wt, corresponding with protein levels. Pull-down samples with LIMP-2 variants Y163C, H171A, K181S and F191S exhibited almost no GCCase activity, confirming the low levels of co-purified GCCase as shown in G. The H171K sample showed slightly reduced activity and the R192S sample showed similar activity compared to the wt sample, confirming these variants ability to bind functional wt GCCase.

4. Both LIMP2 and acid- β -glucosidase reside near or at the membrane, and the discussion of the structure in context of the membrane needs to be more clear. What are the two lines of membrane at the top and bottom of Figure S6A? Does Figure S6B make sense in the context of acid- β -glucosidase interaction with the membrane (ie. the N-terminus)?

We apologize for the missing clarity and adapted Figure S6A and B accordingly.

5. Discussions of dimerization of both proteins should be further supported by additional data. Is the LIMP-2 dimer presented even seen in solution or could it possibly be an artifact of crystallography? Also, the authors claim that “GCase dimerizes in the presence of SAPC as well as some artificial ligands, which were suggested to be beneficial for GCase activity.” There is also literature precedent for acid- β -glucosidase monomerization in the presence of SapC (Gruschus et al. 2015, *Biochem. Biophys. Res. Commun.* “Dissociation of glucocerebrosidase dimer in solution by its co-factor, saposin C”). Additional support for the role of dimerization (or not) for LIMP2 or acid- β -glucosidase would strengthen the study.

We agree with the reviewers that the open questions with regard to dimerization of both GCase and LIMP-2 need to be more critically discussed. In line with our discussion of above comments, we hope that with the additional data included into the manuscript, we can convincingly show that we only observed both GCase and LIMP-2 as monomers or as a heterodimer throughout our study, even without PMbs present. We would also like to refer to our previous publication [Ref Adv Sci hier], where we initially presented the sLIMP-2-derived purification approach. In that study, we show in more detail that the sLIMP-2 construct is purified as a monomer, running as a single peak in SEC. We did not observe a sLIMP-2 dimer in solution. Our structure suggests that dimerization of LIMP-2 as published in the study of Conrad et al. is impossible when GCase is bound, but we do not want to question the findings of that study. Also, we do not see a contradiction to our study as the Conrad study proposed LIMP-2 dimers at the cell surface of epithelial cells, while the GCase/LIMP-2 transport complex forms in the ER and resides within the cell. As we only focused on the luminal domain of LIMP-2 we might miss some effects stemming from the transmembrane regions or the C- and N-terminal ends. For GCase dimerization was also shown to be concentration dependent (Gruschus et al. 2015, *Biochem. Biophys. Res. Commun.* “Dissociation of glucocerebrosidase dimer in solution by its co-factor, saposin C”). The same might hold true for LIMP-2. Maybe we never reached the concentration at which LIMP-2 would form such dimers.

With regard to GCase dimerization, we have modified the manuscript to shed more light on GCase dimerization and its biological relevance. As mentioned before, we have not observed GCase dimerization under our experimental conditions. We hope that we could address this open question better with the new data and discussion included in the revised manuscript.

Reviewer #3 (Remarks to the Author):

A biologically-relevant structure of the GCCase-LIMP2 complex would aid in the design of small molecules or peptides or proteins that could be useful for the treatment of Gaucher disease and/or the form of Parkinson's that is linked to GD. However, it is unclear from this paper that the structure presented is biologically relevant. There are several issues that need to be resolved:

1. The critical issue is whether this tetrameric complex is SELECTED due to the affinity of two previously characterized antibodies or whether they are merely tags that allow the structure to be determined. Related is the issue of how the COMBINATION of nanobodies effect the structure of GCCase and its binding to LIMP2. The bioRxiv manuscript that reported the production and characterization of the two nanobodies show potential effects on GCCase activity and GCCase thermal stability...both structure-driven. HOWEVER, the combination of N1 and N6 is NOT reported. This is critical information

We thank the reviewer for this critical point, which is in line with comments from other reviewers and addressed it experimentally. Part of this data was also included into the main Figures or the supplement of the revised manuscript. Here, we cannot show all data, as it is part of the other manuscript by Dal Maso et al.,

1. We used SEC to show simultaneous binding of both Nbs to GCCase (Figure below, A and new Figure S1C). We can observe a continuous shift of the peak after the addition of one or two Nbs to GCCase.
2. We evaluated the activity profile for GCCase in the presence or absence of one or two Nbs. We can now show that GCCase retains an increased activity after the addition of both Nbs (Figure below, B).
3. Performed a thermal shift assay and could see additive beneficial effects if both Nbs were added to GCCase as the melting temperature for GC increased substantially (Figure below C, D).

Figure 1: A) SEC analysis of GCCase alone (blue), Nb1 (yellow) and Nb6 (green), as well as a combination of GCCase with individual Nbs (Nb1 red, Nb6 orange) or both (black). B) activity assay of GCCase alone or in

combination with one or two Nbs. Conduritol- β -epoxide (CBE) was used as a negative control. C) List of the melting points (three replicates) from a thermal shift assay of GCase (upper row) or GCase in combination with CBE, Nb1, Nb6 or both Nb1+Nb6 at neutral and lysosomal pH. D) Bar graph of the delta in melting temperature in comparison to GCase alone.

Additionally, we gathered data that suggests the presence of a 1:1 stoichiometric complex in our GCase/LIMP-2 co-expression system that is not induced by the presence of the PMbs.

1. We now added low resolution data on the GCase/LIMP-2 complex that we deduced from negative stain images of the complex without PMbs. Unfortunately, the GCase/LIMP-2 complex alone did not yield enough symmetry breaking features for a high-resolution alignment from cryo-TEM images. Thus, we had to include the PMbs to (i) obtain an individual SEC peak and (ii) to introduce features for image alignment. The negative stain reconstruction of the GCase/LIMP-2 complex yields a low-resolution structure that fits in size and shape to a 1:1 stoichiometry after rigid docking of the atomic model obtained from the cryo-TEM dataset (Figure below and new Figure S1A, B). Thus, we anticipate that we do not have a substantial amount of other complex stoichiometries in our sample than the reported 1:1.

Figure S1A, B: A) negative stain electron microscopic image and class sum images from the GCase/LIMP-2 protein complex without PMbs. B) 3D map of GCase/LIMP-2 reconstructed from negative stain images (upper row). Rigid docking of the atomic model of GCase and LIMP-2 as reconstructed from the cryo-TEM experiments (lower row).

2. By the design of our purification approach, all purified GCCase must be bound to sLIMP-2. Therefore, we expect all GCCase detected in SEC fractions to be part of a GCCase/sLIMP-2 complex. Even without the presence of PMBs, we observe only a single peak in the SEC chromatogram that contains GCCase, suggesting that only one stoichiometry is present for the GCCase/LIMP-2 complex. Elution of the complex peak close to the sLIMP-2 monomer peak during SEC suggest a particle size increase that is more likely corresponding to a single GCCase molecule. If the initially purified complex would have 2:1 stoichiometry, competitive binding of the PMBs would yield free GCCase as an additional peak in the chromatogram, which we did not observe. In line with another reviewer comment, we have also added GCCase activity data in SEC fractions (new Figure 1D), confirming that A: GCCase in the purified complex is bioactive and B: no free GCCase can be detected after PMBs addition.

Figure 1B-D in revised manuscript: B: SEC chromatograms of sGCCase/LIMP-2 samples with and without PMBs added. Dashed line: no PMBs added. Solid line: PMBs added. Peaks are annotated with corresponding proteins. Addition of PMBs lead to emergence of novel PMb monomer peaks but also caused a shift of the sGCCase/LIMP-2 peak, indicating formation of a sGCCase/LIMP-2/PMb complex. C: Analysis of SEC fractions (total protein and western blot) confirming presence of all four proteins of interest in fractions 5-8, corresponding to the tetramer peak seen in B (MBP = maltose binding protein as part of the PMbs). Fraction 6 is highlighted in red as it was used for further structural analyses of the protein complex. D: Distribution of GCCase activity across SEC fractions 5-10 of GCCase/sLIMP-2 samples with and without PMBs. as seen in the SEC chromatogram (B) and protein analysis (C), binding of PMBs leads to a shift of active GCCase towards earlier fractions, confirming that GCCase retains bioactivity after binding of PMBs.

3. In an earlier manuscript (Dobert et al., 2024; PMID: 38666485), we expressed GCCase alone or in combination with LIMP-2 and performed SEC experiments. GCCase alone (Figure 3E, F; gray dashed line) runs in earlier fractions than co-expressed with LIMP-2. Together with data from another paper that indicates GCCase dimer formation in a concentration dependent manner, we might not reach the critical concentration for GCCase monomers to form dimers when co expressed together with LIMP-2 (Gruschus JM et al., 2015; PMID: 25600808). Additionally, we cannot exclude an impact of the His-tag attached to the GCCase when expressed alone. For endogenous GCCase and GCCase co-expressed with LIMP-2 no prominent GCCase peak could be detected in the fractions where the His-tagged GCCase eluted (Figure below). For untagged GCCase (Imiglucerate), as we used for the binding experiments with the Nbs, we did not observe a peak in SEC that would correspond to a GCCase dimer (Figure 1 above).

Figure 3 E, F from Dobert et al., 2024: E) SEC and F) Western blot analysis of GCCase/LIMP-2 (blue line), LIMP-2 alone (black line) and His-tagged GCCase alone (gray dashed line).

2. The binding of saponin C to GCCase is NOT UNDERSTOOD (models have not been tested) at a structural level....so any interpretations of how LIMP-2 binding and sapC binding are related should NOT be included.

We thank the reviewer for this important point. We took the comment as challenge to investigate the interaction of GCCase with SapC using experimental data. Although we cannot report a structure, we used cross-linking experiments in combination with mass spectrometry to assess binding regions of SapC on GCCase. Our cross-linking experiments provide initial experimental data about the binding of SapC to GCCase, which we included as a new supplementary Figure S8 to the revised manuscript. The reported binding sites are not in accordance with the data reported based on in silico docking experiments and we therefore removed these data. We omitted SapC data from the main figures to also streamline the story on GCCase/LIMP-2. In the future we aim to unravel the structural universe of GCCase in the lysosome, including structural investigations of GCCase in complex with SapC.

3. I would like to see an experimental test of the relevance of the structure (as opposed to a rationalization based on existing data that is itself not compelling). Demonstrate that elimination of a key interaction (the His 171 "switch" for example) can disrupt transport to the lysosome.

We thank the reviewer for this suggestion, which has been made by the other reviewers as well. We have conducted a set of new experiments to reinforce the biological relevance of the presented structure and characterized GCCase and LIMP-2 variants.

We generated a total of three GCCase and six sLIMP-2 variants, for which we performed experiments to assess expression, activity and capability to form a GCCase/LIMP-2 complex. Along the previously mentioned D409H variant, we included two more disease-associated GCCase variants residing in the interface region (E388K and R395C). We further generated two SNP-derived sLIMP-2 variants identified in the interface (Y163C and F191S), as well as four artificial sLIMP-2 variants to target amino acid interactions identified by our structural analysis (H171A, H171K, K181S and R192S). The K181S variant targets the salt bridge formed with Asp409 from GCCase and is thus complementary to the D409H variant. All variants were co-expressed in HEK 293F cells along their respective wt interaction partner. We obtained whole-cell-lysates to assess expression and activity of the variants and performed Ni-NTA pulldown experiments from the cell culture supernatant to purify GCCase/sLIMP-2 variant complexes via the LIMP-2 attached 10xHis-tag for activity measurements.

To address the specific question on pH induced dissociation of the GCCase/LIMP-2 complex, we included the previously reported mutations for His171 of LIMP-2 (Zachos et al.,). We can now report, that His171 is most likely important for proper folding of the LIMP-2 interaction interface with GCCase. Even at neutral pH we did not observe complex formation when we changed His171 to alanine (H171A). In case we included a positively charged and more bulky lysine, GCCase activity was reduced

to ~60% after Ni-NTA pulldown. Unfortunately, we cannot make a statement on any pH induced changes of the protein complex at this point and have to postpone the question to a later study. It needs to be answered in a more comprehensive follow-up investigation, which should also account for other lysosomal interaction partners such as SapC.

We have included our data on the GCase and LIMP-2 variants in a new Figure 5 and added a new chapter to our manuscript presenting our findings. We have now identified key residues important for complex formation. We think that these experiments significantly substantiate and improve our study and thank the reviewer again for this thoughtful suggestion.

Figure 5: Effect on amino acid substitutions within the GCase/LIMP-2 interface. **A:** Overview of GCase/LIMP-2 interface (orange: GCase, blue: LIMP-2). Analyzed variants (red: GCase; dark blue: LIMP-2) are highlighted and annotated. **B:** Detailed views of each GCase and LIMP-2 variant generated for this study. The targeted amino acid is highlighted and surrounding or interacting amino acids of the interaction partner are shown and labeled. Potential hydrogen bonds (green) and salt bridges (yellow) are shown as a dashed line. **C, D:** Western Blot analysis of whole-cell lysates of HEK 293F cells co-expressing GCase variants with wt sLIMP-2 (C) and sLIMP-2 variants with wt GCase (D). All variants were expressed, although some variants showed reduced expression levels and also affected expression of their co-expressed wt interaction partner. **E:** GCase activity in whole cell lysate samples from C, D. The GCase variants exhibited reduced activity compared to the wt. Some sLIMP-2 variants negatively impacted GCase activity, corresponding to their effect on GCase expression shown in D. **F, G:** Western Blot analysis of Ni-NTA pulldown from culture medium of HEK 293F cells co-expressing GCase variants with wt sLIMP-2 (F) and sLIMP-2 variants with wt GCase (G). R395C and D409H GCase variants showed strongly reduced purification yield of both sLIMP-2 and GCase while the E388K variant behaved similar to the wt. **H:** GCase activity in Ni-NTA pulldown samples from F, G. Activity of E388K GCase was slightly reduced despite similar-to-wt protein levels observed in F. Activities of R395C and D409H GCase were strongly reduced compared to the wt, corresponding with protein levels. Pulldown samples with LIMP-2 variants Y163C, H171A, K181S and F191S

exhibited almost no GCCase activity, confirming the low levels of co-purified GCCase as shown in G. The H171K sample showed slightly reduced activity and the R192S sample showed similar activity compared to the wt sample, confirming these variants ability to bind functional wt GCCase.

4. assay of GCCase activity in the presence of taurocholate and the absence of a membrane or micellar surface are not relevant. Taurocholate is likely an allosteric activator itself!

We strongly agree with the reviewer's comment regarding the relevance of typically performed GCCase activity assays. The reviewer is completely right that cleavage of a synthetic substrate in absence of membranes does not necessarily reflect the capability of GCCase to process native substrates within the lysosome. For this study, we utilized GCCase activity assays to locate GCCase within samples and SEC fractions in support of our western blot data. We also used it to judge whether the purified enzyme retained bioactivity to rule out any non-physiological interactions caused by misfolding, which is not visible in western blot analysis. The purpose of activity measurements in this context was only to assess basic enzymatic function of GCCase. The activity data for the purified GCCase/sLIMP-2 complex was omitted from the initial the manuscript by accident and we have added it to the revised version (Figure 1D). Further, our newly added experiments leveraged GCCase activity measurements to assess the integrity of GCCase variants in order to gauge whether the introduced amino acid changes had direct effects on enzyme function.

In this study, we did not investigate or quantify any effect of a binding partner of GCCase on its enzyme activity. This was not the goal of this study. But to reiterate our agreement with the reviewer's comment, we would like to refer to our previous study, where we did explore possible effects of LIMP-2 on GCCase activity (Dobert et al., 2024). In the referenced study, we consciously omitted taurocholate from the activity assay when testing effects of LIMP-2, exactly to eliminate any confounding activating effect of the taurocholate itself. We also substantiated our findings using live cell lysosomal activity measurements to more accurately reflect GCCase activity under physiological conditions.

5. The fact that the active site residues are not well-resolved in the complex argue against allosteric activation as the mech of LIMP2 binding

As touched on in the previous comment, this study did not investigate any allosteric activation effects of LIMP-2 or GCCase nanobodies. The potential allosteric effects of these interaction partners are part of other studies, which have been previously published or are in the process of publishing (Ref here Dobert et al. Dal Maso et al. preprint). Nonetheless, we are thankful for this critical comment and agree that the mechanisms of allosteric activation that have been described previously for LIMP-2 and the nanobodies cannot be explained by the structure presented in this study. We think that follow-up studies are necessary to pinpoint the molecular mechanisms by which activation of GCCase is achieved, which would also be tremendously beneficial for GCCase-targeted drug design. It will be our goal in the future to determine a GCCase structure with a native substrate bound to understand the impact of the active site residues and surrounding loops for enzymatic activity.

I strongly feel that the impact of this structure will be minimal if presented in its current form BUT would be significant if bolstered with a few mutagenesis experiments....

We thank the reviewer for the critical input and suggestions for improvement and hope that the new experiments included in our revised manuscript improve the relevance and robustness of our study and render it suitable for publication.

Rebuttal letter

Reviewer #1 (Remarks to the Author):

The authors have adequately addressed my comments and concerns.

We thank the reviewer!

Reviewer #3 (Remarks to the Author):

I am very happy (and impressed) with the additional data and discussion added by the authors. This kind of response is unfortunately rare. I am in favor of publication

We thank the reviewer and take this comment as a compliment especially for our PhD student who put the data together.

Reviewer #4 (Remarks to the Author):

I have gone over the MS by Dobert et al. and find that the authors have done a commendable job in addressing the concerns raised by the three first-round reviewers. The new data on Saposin C are suggestive but not compelling, but this is included in the Supplementary info and does not affect the main paper. The authors state that the complex with SaposinC will be addressed in future studies, and I agree. The MS is well written and well presented. The central conclusions are sound and bolstered by the new mutagenesis data.

We thank the reviewer and feel that the rather suggestive SapC data will provide an excellent starting point for follow up experiments.